

# Controls on dense water formation along the path of the North Atlantic subpolar gyre

Oliver J. Tooth [1,2], Helen L. Johnson [1], and Chris Wilson [3]

[1]University of Oxford, Oxford, United Kingdom
[2]National Oceanography Centre, Southampton, United Kingdom
[3]National Oceanography Centre, Liverpool, United Kingdom

**Correspondence:** Oliver J. Tooth (oliver.tooth@earth.ox.ac.uk)

**Abstract.** The North Atlantic Subpolar Gyre (SPG) plays a fundamental role in the global climate system through the formation of dense North Atlantic Deep Water (NADW) as part of the Atlantic Meridional Overturning Circulation. Observations show pronounced decadal variability in SPG water mass properties; however, it remains unclear to what extent such thermohaline changes impact the formation of dense water. Here, we explore the mechanisms governing dense water formation along the path of the SPG using Lagrangian water parcel trajectories in an eddy-rich ocean sea-ice hindcast. We show that neither the rate of transformation of water parcels across density surfaces nor their thermohaline properties on arrival into the eastern SPG are rate-limiting factors governing dense water formation. Instead, the total amount of dense water formed during transit around the SPG can be skilfully predicted based solely on the volume transport of light, upper limb waters arriving into the eastern SPG via the branches of the NAC. This relationship between upper limb volume transport and dense water formation emerges since the SPG boundary current is long enough for all upper limb thermal anomalies to be damped during transit. Multi-decadal subpolar overturning variability in density-space is therefore closely related to the strength of the SPG, such that a stronger SPG circulation following persistent positive phases of the North Atlantic Oscillation results in greater NADW formation along-stream. Our findings emphasise the coupling between the SPG and overturning circulations and underscore the importance of monitoring the state of the SPG for both decadal and longer-term climate predictions.

## 1 Introduction

Observations and ocean reanalyses indicate that the upper subpolar North Atlantic Ocean (SPNA) exhibits pronounced thermohaline variability on decadal timescales (Curry et al., 1998; Bersch, 2002; Bersch et al., 2007; Lozier and Stewart, 2008; Holliday et al., 2018, 2020; Fu et al., 2020), with significant implications for both regional and global climate (e.g., Zhang et al., 2019; Kim et al., 2020). On a regional scale, the northward propagation of upper ocean thermohaline anomalies from the SPNA into the Nordic Seas yields predictable climate impacts on surface air temperatures over northwest Europe (Collins and Sinha, 2003; Keenlyside et al., 2008; Li et al., 2013; Årthun et al., 2017), Arctic sea ice extent (Zhang, 2015; Yeager et al., 2015) and the rates of Greenland glacial melting (Straneo and Heimbach, 2013). The impacts of North Atlantic multi-decadal SST variability also extend far beyond the regional scale, including through hemispheric teleconnections which contribute to changes in West African and Indian summer monsoon rainfall (Feng and Hu, 2008; Goswami et al., 2006; Luo et al., 2011;



Martin and Thorncroft, 2014; Martin et al., 2014) and Pacific decadal climate variability (Zhang et al., 2007; d'Orgeville and Peltier, 2007; Zhang et al., 2019).

Both observational and numerical modelling studies have repeatedly identified low-frequency subpolar ocean dynamics as the principal source of high-latitude upper ocean thermohaline variability (Årthun and Eldevik, 2016; Desbruyères et al., 2015; Grist et al., 2010; Robson et al., 2016; Yeager and Robson, 2017; Chafik et al., 2023). In particular, multi-decadal changes

in the subpolar gyre (SPG) and overturning circulations (Yeager et al., 2021; Kim et al., 2024), excited by fluctuations in the North Atlantic Oscillation (NAO; Hurrell 1995; Marshall et al. 2001), have been shown to modulate the poleward heat transport by warm and saline subtropical waters into the eastern SPNA (Jacobs et al., 2019; Desbruyères et al., 2013). Two important examples are the 1990s and 2010s when persistent positive phases of the NAO induced a delayed warming and salinification of the upper eastern SPG (Desbruyères et al., 2021; Fu et al., 2020; Chafik et al., 2023), which propagated downstream into

the Nordic Seas (Fan et al., 2023; Passos et al., 2024). This potentially predictable influence of subpolar ocean dynamics on the North Atlantic climate system is further underscored by initialised decadal predictions (Marotzke et al., 2016; Smith et al., 2019; Boer et al., 2016), which show a strong sensitivity in retrospective forecast skill to the initialised subpolar ocean state (e.g., Robson et al., 2012; Msadek et al., 2014; Hermanson et al., 2014; Yeager and Robson, 2017).

To date, very few studies (e.g., Passos et al., 2024) have investigated the extent to which the arrival of such anomalies persist

to impact the formation of dense water masses downstream. On the one hand, we might expect that upper ocean thermohaline anomalies will feed back onto the strength of the subpolar overturning circulation by impacting the efficiency of diapycnal water mass transformation and, therefore, the production of North Atlantic Deep Water (NADW) downstream. Indeed, this view forms the basis of the salt-advection feedback (Stommel, 1961; Rahmstorf, 1996; de Vries and Weber, 2005), in which a weakened Atlantic Meridional Overturning Circulation (AMOC) transports less salt into the SPNA, thereby reducing NADW

formation through increasing stratification, which further weakens the AMOC. However, this view is at odds with a number of recent observational studies (Fu et al., 2020; Fraser and Cunningham, 2021; Mercier et al., 2024; Koman et al., 2024), which show that the strength of subpolar overturning has remained relatively stable in spite of the large-scale thermohaline variability observed throughout the SPNA during recent decades. Fu et al. (2020) reconcile this by suggesting that only a weak coupling exists between upper limb thermohaline anomalies and the magnitude of subpolar dense water formation on multi-decadal

timescales. However, precisely what controls the amount of dense water formed along the path of the SPG and its relationship to subpolar overturning variability on multi-decadal timescales remains poorly understood.

When exploring the downstream evolution of upper ocean thermohaline anomalies in the SPNA, studies adopting the traditional Eulerian frame of reference typically use lagged correlation analysis (Holliday et al., 2008; Årthun and Eldevik, 2016; Årthun et al., 2017; Fan et al., 2023), which relies upon the coherent propagation of signals downstream. In reality, how-

ever, thermohaline anomalies are communicated over a diverse range of advective timescales owing to the dispersive nature of subpolar circulation pathways (e.g., Yashayaev and Seidov, 2015), which often convolve water masses from many different sources. In this study, we overcome this challenge by adopting a Lagrangian approach to investigate the controls on dense water formation along the path of the SPG. By evaluating Lagrangian water parcel trajectories in an eddy-rich ocean sea-ice





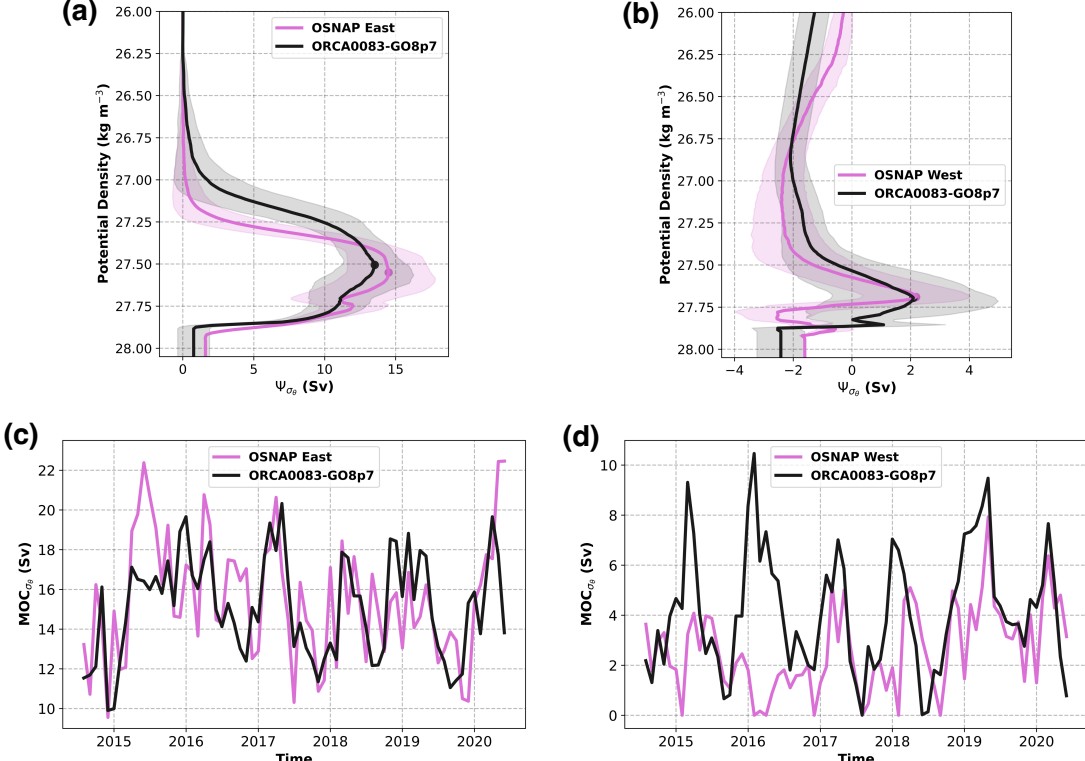

**Figure 1.** Time-mean (2014-2020) Eulerian diapycnal overturning stream functions at (a) OSNAP East and (b) OSNAP West calculated using the ORCA0083-GO8p7 hindcast (black) and OSNAP observations (purple). The shaded regions represent $\pm 1$ (monthly) standard deviation in the Eulerian overturning stream function. Time series of maximum Eulerian diapycnal overturning at (c) OSNAP East and (d) OSNAP West calculated using the ORCA0083-GO8p7 hindcast (black) and OSNAP observations (purple).

hindcast, we are able to trace the evolution of upper limb thermohaline anomalies arriving into the eastern SPNA and directly

assess their influence on the formation of NADW during their transit of the SPG.

This study is organised as follows. In Section 2, we introduce the eddy-rich ocean sea-ice hindcast, Lagrangian particle tracking experiments and both the Eulerian and Lagrangian diagnostics used in our analysis. Section 3.1 explores the nature of dense water formation along the path of the SPG, including validation against OSNAP observations. We investigate the sources of variability in along-stream dense water formation in Section 3.2. In Section 3.3, we propose a linear model to

skilfully predict along-stream dense water formation. Section 3.4 explores the two dense water formation pathways circulating cyclonically around the SPG. Finally, in Section 3.5, we assess the role of remote buoyancy forcing in driving decadal variations in dense water formation along the path of the SPG. The study concludes with a critical discussion and summary of our main findings and their wider implications for observing subpolar AMOC variability in Section 4.





## 2 Methods

### 2.1 Ocean General Circulation Model

To investigate the variability of dense water formation along the path of the subpolar gyre, we use output from the eddy-rich, global ORCA0083-GO8p7 numerical ocean model configuration (Megann et al., 2022; Archibald et al., 2025). This uses the Nucleus for European Modelling of the Ocean (NEMO) ocean circulation model version 4.0.4 (Madec et al., 2019) implemented in the UK Global Ocean (GO) version 8 configuration. This is a pre-release configuration of the Global Ocean and Sea Ice (GOSI) version 9 (Guiavarc'h et al., 2025) developed by the UK Joint Marine Modelling Programme in preparation for CMIP7.

The NEMO ocean model is discretised on an Arakawa C-grid with a nominal 1/12° resolution (equivalent to ∼8 km in the subtropical North Atlantic and ∼4 km in the Arctic). The extended version of the quasi-isotropic ORCA12 orthogonal tri-polar grid (eORCA12) is used, with poles located on land in Canada, Siberia and Antarctica. In the vertical, the model uses 75 unevenly spaced z*-partial-step coordinate levels with unperturbed depth increments ranging from 1 m to 250 m. The depth increment of the grid cells at each vertical level varies through time due to the implementation of a non-linear free surface (Madec et al., 2019). Within the NEMO framework, the NEMO Ocean Engine (NEMO-OCE) is coupled to the SI$^3$ sea-ice model (Vancoppenolle et al., 2023; Blockley et al., 2024). For a comprehensive description of the ORCA0083-GO8p7 model configuration users are referred to Guiavarc'h et al. (2025).

The ORCA0083-GO8p7 integration is initialised from rest with temperature and salinity from the climatology of an Argo-based observational objective analysis (EN4; Good et al., 2013) covering 1995-2014. The model is forced by the Japanese 55-year atmospheric reanalysis (JRA55-do; Tsujino et al., 2018) for the period 1958-2021. We disregard the initial 18 years of the integration when model adjustment is largest and make use of the monthly-mean velocity and tracer fields output between 1975-2021.

To assess the fidelity of the subpolar ocean circulation in the ORCA0083-GO8p7 hindcast, we compare the simulated diapycnal overturning to OSNAP observations between 2014-2020. Figures 1a-b show the time-mean Eulerian diapycnal overturning stream functions at the OSNAP East and OSNAP West sections calculated using both the model and observations. Overall, we find good agreement between the modelled and observed overturning stream functions in density space at the OSNAP array. At OSNAP East, the maximum of the time-mean diapycnal overturning stream function in the model (13.5 ± 2.3 Sv at 27.51 kg m$^{-3}$) is slightly weaker than observed (14.5 ± 3.0 Sv at 27.55 kg m$^{-3}$). However, this is primarily due to the weaker time-mean net northward transport across the section in the model (0.8 ± 1.1 Sv) compared to the 1.6 Sv imposed in the OSNAP observational calculation. Figure 1b shows that there is also close agreement between both the magnitude and isopycnal of maximum diapycnal overturning in the model (2.1 ± 2.8 Sv at 27.70 kg m$^{-3}$) and observations (2.2 ± 1.8 Sv at 27.69 kg m$^{-3}$) at OSNAP West. This is particularly encouraging, given that many eddy-rich models considerably overestimate the time-mean strength of diapycnal overturning in the Labrador Sea (e.g., Petit et al., 2023; Markina et al., 2024). Once we account for the larger net southward flow across the OSNAP West section in the model (-2.4 ± 0.8 Sv) compared with that imposed in





the OSNAP observational calculation (-1.6 Sv), we do find that the modelled diapycnal overturning is slightly stronger than observed.

Figures 1c-d show the time series of diapycnal overturning strength at OSNAP East and OSNAP West determined by

calculating the maximum of each monthly overturning stream function. Although we find a significant correlation ($r = 0.55$, $p < 0.01$) between the modelled and observed diapycnal overturning strength at OSNAP East, it is clear that the model (monthly SD = 2.6 Sv) underestimates the monthly overturning variability captured in observations (monthly SD = 3.0 Sv). In contrast, at OSNAP West, we find a much weaker correlation ($r = 0.27$, $p < 0.05$) since the observed diapycnal overturning strength is less variable than found in the model, especially on seasonal timescales. The stronger overturning seasonality in the model is due

to the presence of warmer (+ 0.14°C) and saltier (+ 0.1 g kg$^{-1}$) waters in the western Labrador Sea compared with OSNAP observations, which experience less density compensation between wintertime cooling and year-round freshening (Zou et al., 2020; Bebieva and Lozier, 2023).

In addition to reproducing much of the observed strength and monthly variability of overturning along the OSNAP array, we also find reasonably good agreement between the modelled (6.5 ± 1.0 Sv) and observed (5.8 ± 0.7 Sv; Østerhus et al., 2019)

overturning strength at the Greenland-Scotland Ridge (1995-2015). Given our focus on the formation of dense water along the boundary current of the SPG in this study, we also highlight the close agreement between the time-mean top-to-bottom strength of the East Greenland Current (-33.8 ± 2.7 Sv) in the model and that observed by Daniault et al. (2016) (-33.1 ± 2.6 Sv) between 2002-2012 along the OVIDE section. However, in the Labrador Sea, the model (-27.4 ± 5.8 Sv) slightly underestimates the time-mean (1997-2014) strength of the DWBC at 53°N as reported by Zantopp et al. (2017) (-30.2 ± 6.6

Sv where $\sigma_\theta \geq 27.68$ kg m$^{-3}$).

In spite of these differences, we consider the broad overall agreement between the strength of both the SPG and diapycnal overturning circulations simulated in the ORCA0083-GO8p7 hindcast and that observed along trans-basin arrays as sufficient justification for using this model to investigate the nature of dense water formation along the path of the SPG.

## 2.2 Lagrangian Particle Tracking

We evaluate the Lagrangian trajectories of virtual water parcels advected by the time-evolving velocity fields of the ORCA0083-GO8p7 hindcast using TRACMASS version 7.1 (Aldama-Campino et al., 2020). TRACMASS belongs to the inaugural family of Lagrangian particle tracking tools, which allow users to quantify the volume transport pathways of steady, incompressible flow field by modelling water parcel trajectories as individual stream tubes (e.g., Blanke and Delecluse, 1993; Döös, 1995). Here, we use the stepwise stationary scheme, which divides the duration between successive monthly-mean velocity fields into

100 intermediate time steps during which volume transports are assumed constant. We calculate purely advective water parcel trajectories without attempting to parameterise the effects of vertical convective mixing in the surface mixed layer. This is because Tooth et al. (2023b) showed that introducing random vertical displacements along water parcel trajectories inside the surface mixed layer did not influence the strength and variability of along-stream diapycnal transformation.

The Lagrangian experiment used in this study evaluates the trajectories of water parcels sampling the full-depth northward

transport across a subsection of the OSNAP East array extending from the Reykjanes Ridge (RR, 30°W) to the Scottish Shelf





(SS). We choose this subsection of the array to focus our analysis on the waters which flow northward across OSNAP East in the northern, central and southern branches of the NAC and to avoid sampling the recirculating upper limb waters which return northward in the Irminger Current (see Fig. 2). Water parcels are initialised every month for 456 consecutive months between 1975-2012. Water parcels are assigned to each grid cell along the model-defined OSNAP East array (RR-SS) in proportion to the northward volume transport across the grid cell face. Each water parcel represents a volume transport $\leq 2.5$ mSv to ensure that a sufficiently large number of water parcels are initialised to calculate robust Lagrangian diagnostics (Jones et al., 2016).

In total, more than 12.5 million water parcels are advected forwards-in-time using monthly mean velocity fields for a maximum of 9-years to determine their future trajectories. Water parcel trajectories are terminated on reaching the maximum advection time ($\tau_{max}$) or upon meeting any one of the following geographical criteria (Fig. 2): (i) crossing (southward) a subsection of the OSNAP West array (53°N) (ii) crossing the Greenland-Scotland Ridge, (iii) crossing the Davis or Hudson Straits or (iv) crossing 51°N. Figures 3a-c show that the 9-year maximum advection time is sufficient to capture the SPG circulation because the accumulated volume transports reaching OSNAP West and the Greenland-Scotland Ridge have stabilised within this period. The location, conservative temperature and absolute salinity along each water parcel trajectory are calculated through linear interpolation using the monthly-mean model tracer fields. The potential density referenced to the sea surface ($\sigma_\theta$) is calculated along each trajectory using the TEOS-10 equation of state (McDougall et al., 2012) as implemented in the ORCA0083-GO8p7 hindcast (Megann et al., 2022).

### 2.3 Lagrangian Diagnostics

To quantify the amount of dense water formed along the path of the North Atlantic SPG, we calculate the Lagrangian diapycnal overturning stream function (Tooth et al., 2023a), $F(\sigma_\theta, t)$, in density-coordinates using only the subset of water parcel trajectories initialised at time $t$ which transit from OSNAP East (NAC) to OSNAP West (53°N) within the $\tau_{max} = 9$-year maximum advection period (i.e., $0 < \tau_{out} \leq \tau_{max}$ where $\tau_{out}$ is the transit time for each water parcel). We focus on this particular subset of water parcels since the overwhelming majority of water parcels which exit via the southern boundary at 51°N or remain inside the domain following 9 years of advection are already contained within the lower limb on flowing northward across the OSNAP East section and therefore are not involved in dense water formation. The Lagrangian diapycnal overturning stream function for each monthly ensemble of $N(t)$ water parcels initialised across the OSNAP East section at time $t$ is calculated following Tooth et al. (2023a, b):

$$F(\sigma_\theta, t) = \sum_{\sigma_\theta^* = \sigma_{min}}^{\sigma_\theta} V_{NAC, \sigma_\theta^*}(t) - V_{53°N, \sigma_\theta^*}(t + \tau_{out}) \tag{1}$$

where $V_{NAC, \sigma_\theta^*}$ and $V_{53°N, \sigma_\theta^*}$ represent the absolute volume transport distributions of SPG water parcels in density-coordinates on their initial northward (NAC) and final southward (53°N) crossings of the OSNAP array.

In order to investigate the downstream evolution of thermohaline anomalies, we additionally define the volume-weighted mean of a specified quantity $q$ (e.g., potential density) for each monthly ensemble of $N(t)$ water parcels at some time $\tau$





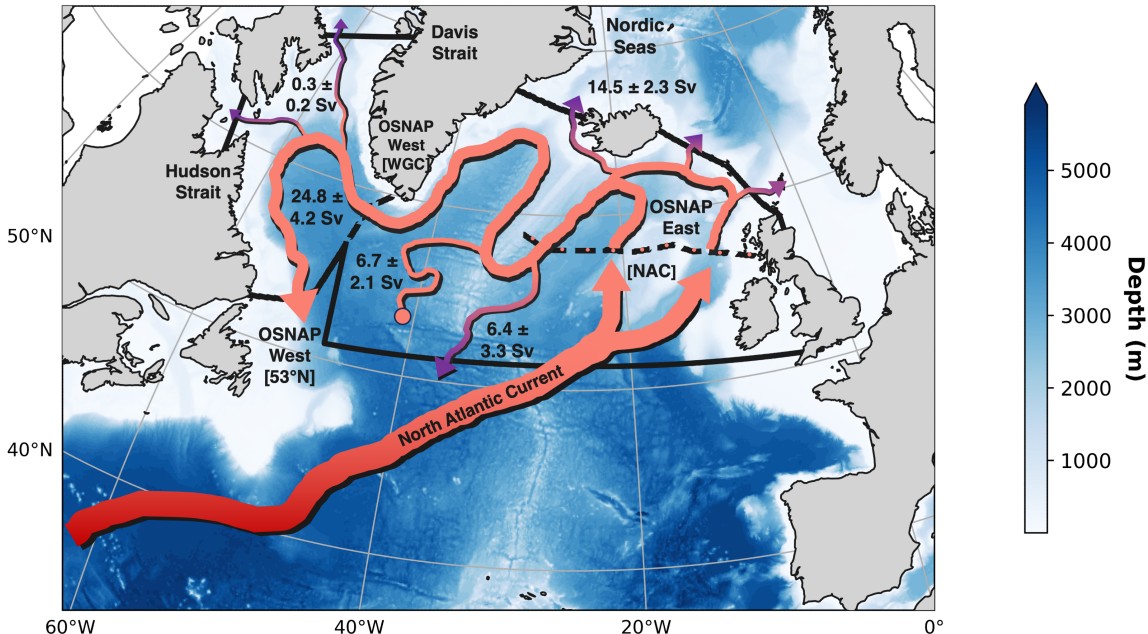

**Figure 2.** Schematic of the North Atlantic Subpolar Gyre (SPG) circulation and Lagrangian experiment domain (solid black lines). Water parcels are initialised northward across the subsection of the OSNAP East array extending from the Reykjanes Ridge ($\sim$31$^\circ$W) to the Scottish Shelf (NAC, black dashed line with orange markers). The SPG pathway (orange) contains all water parcels which flow southward across the OSNAP West array between the Labrador Coast and the basin interior (53$^\circ$N, solid black line) once initialised across OSNAP East and within the 9-year maximum advection period. Purple arrows represent the pathways of water parcels which are terminated on reaching the boundaries of the Lagrangian experiment domain. Water parcels remaining in the Lagrangian experiment domain are represented by the pathway terminating with an orange circle. The time-mean (1975-2012) volume transports conveyed by each pathway are shown in bold.

following their initialisation:

$$\overline{q}(t,\tau) = \frac{\sum_{n \in N(t)} V_n q_n(\tau)}{\sum_{n \in N(t)} V_n} \tag{2}$$

where $V_n$ is the volume transport conveyed by an individual water parcel with index $n$ and $q_n(\tau)$ represents the value of $q$

recorded along its trajectory at time $\tau$ following initialisation, where $t \leq \tau \leq t + \tau_{out}$.

## 2.4 Eulerian Diagnostics

### 2.4.1 Surface-Forced Water Mass Transformation

A major advantage of quantifying the strength of subpolar overturning in density rather than traditional depth coordinates is that it can be directly related to surface buoyancy fluxes and diapycnal mixing through the water mass transformation framework





(Walin, 1982; Marsh, 2000; Speer and Tziperman, 1992; Evans et al., 2023). To quantify the amount of dense water formed by surface buoyancy loss over our Lagrangian experiment domain, we first compute the surface density flux due to the fluxes of heat ($Q_H$, W m$^{-2}$) and freshwater ($Q_{FW}$, kg m$^{-2}$ s$^{-1}$) at the sea surface following Speer and Tziperman (1992):

$$f(x,y,t) = -\frac{\alpha}{C_p} Q_H(x,y,t) + \beta \frac{S(x,y,t)}{1 - S(x,y,t)} Q_{FW}(x,y,t) \qquad (3)$$

where $\alpha$ is the thermal expansion coefficient, $\beta$ is the haline contraction coefficient, $C_p$ is the specific heat capacity of seawater

and $S$ is the sea surface salinity. To calculate the surface-forced diapycnal water mass transformation $H(\sigma_2, t)$ across an outcropping isopycnal surface, we then integrate the surface density flux over the area of each surface density outcrop $\sigma_2^*$:

$$H(\sigma_2,t) = \sum_{\sigma_2^* = \sigma_{min}}^{\sigma_2} \sum_y \sum_x f(x,y,t)\, \Pi(\sigma_2^*(x,y,t))\, \Delta x(x,y)\, \Delta y(x,y) \qquad (4)$$

where

$$\Pi(\sigma_2^*(x,y,t)) = \begin{cases} 1 & \text{for } |\sigma_2^*(x,y,t) - \sigma_2| \leq \frac{\Delta \sigma_2}{2} \\ 0 & \text{elsewhere} \end{cases}$$

The sea surface potential density $\sigma_2$ referenced to 2000 m is computed using model monthly-mean sea surface temperature and salinity fields. The density bin size is given by $\Delta \sigma_2 = 0.02$ kg m$^{-3}$ following Yeager et al. (2021).

### 2.4.2  Definition of Labrador Sea Water

We define Labrador Sea Water (LSW) from the long-term average surface-forced diapycnal water mass transformation calculated over our Lagrangian experiment domain following the methodology of Yeager et al. (2021). The potential density range

of LSW at OSNAP West is determined as the interval over which positive annual mean formation of LSW occurs in the 1975-2012 climatology of $H(\sigma_2, t)$ over the region north of OSNAP West in our Lagrangian experiment domain. In this study, LSW is defined by the density range $\sigma_2 = 37.01$ - $37.11$ kg m$^{-3}$. To account for the lighter composition of LSW in the eastern SPG, we use a modified potential density range $\sigma_2 = 36.95$ - $37.11$ kg m$^{-3}$ to define LSW in the Irminger Sea. We quantify the interior LSW thickness, $\Delta z_{LSW}$, in the Labrador and Irminger Seas by calculating the average layer thickness of LSW defined

by the potential density ranges along both OSNAP West and OSNAP East where the ocean depth exceeds 2000 m.

## 3  Results

### 3.1  Characterising dense water formation along the path of the subpolar gyre

To characterise the nature of dense water formation along the path of the SPG, we begin by describing the circulation pathways of water parcels flowing northward into the eastern SPG between 1975-2012 (Fig. 2).

On average, $52.7 \pm 9.0$ Sv flows northward across the OSNAP East array between the Reykjanes Ridge and the Scottish Shelf via the branches of the NAC. Of this total northward transport, $24.8 \pm 4.2$ Sv ($47.0 \pm 3.1\%$; Fig. 2) circulates around the





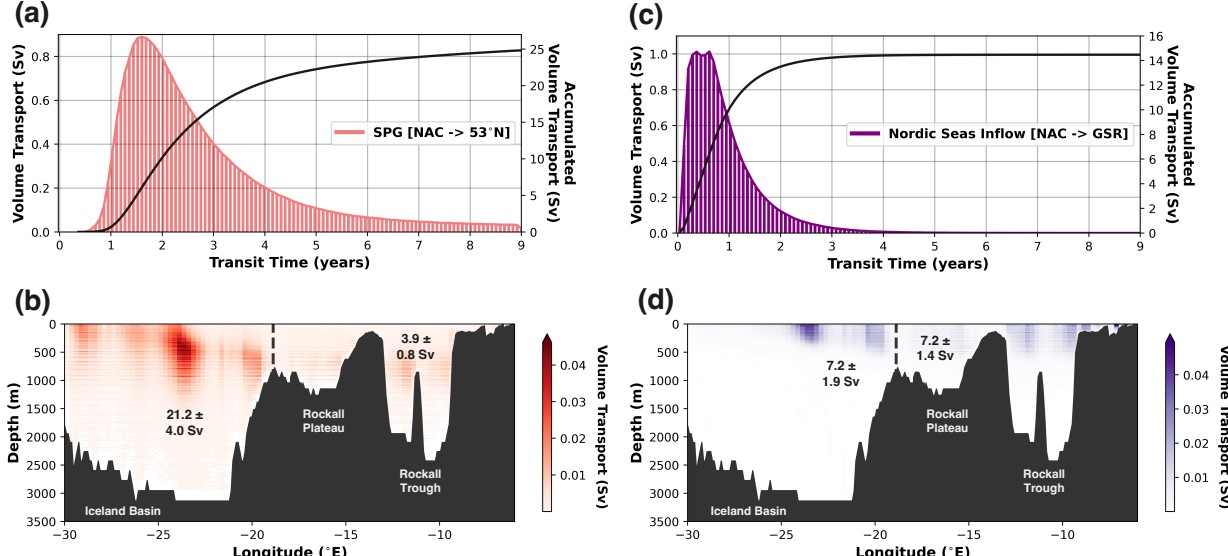

**Figure 3.** (a,b) Volume transport distributions of (a) SPG and (b) Nordic Seas inflow water parcels as a function of the time taken to reach either (a) OSNAP West at 53°N or (b) the Greenland-Scotland Ridge, after initialisation across OSNAP East in the NAC. The solid black lines represent the accumulated volume transport as a function of water parcel transit time. (c,d) Water parcel release locations in the NAC along OSNAP East between the Reykjanes Ridge and the Scottish Shelf for the SPG pathway (c, red) and the inflows to the Nordic Seas (d, purple). The black dashed line at -18.875 °E is used to distinguish between the northward volume transport arriving in the Iceland Basin and that arriving in the Rockall Trough and Plateau. The time-mean volume transport distributions for each pathway are calculated by summing the absolute volume transports conveyed by water parcels in discrete longitude-depth bins, where the bin widths are $\Delta x = 0.25°$E and $\Delta z = 20$ m.

SPG before flowing southwards across the OSNAP West array at 53°N (between the Labrador coast and basin interior), which we shall herein refer to as the SPG pathway. The remaining northward transport across OSNAP East is distributed between pathways crossing the Greenland-Scotland Ridge (27.0 ± 4.5%), the Davis and Hudson Straits (0.5 ± 0.4%) and 51°N (12.4 ± 4.2%), with a small fraction remaining within the SPG interior following 9-years of advection (11.4 ± 2.0%).


Figure 3 shows that the 14.5 ± 2.3 Sv flowing northward into the Nordic Seas is dominated by shallow water parcels, sourced almost equally from the upper 500 m of the central and southern NAC branches, which typically reach the Greenland-Scotland Ridge in less than a year. Interestingly, although this Nordic Seas inflow is larger than observed (Chafik and Rossby, 2019; Østerhus et al., 2019), we find reasonable agreement between the modelled (6.5 ± 1.0 Sv) and observed (5.8 ± 0.7 Sv; Østerhus et al., 2019) overturning strength at the Greenland-Scotland Ridge (1995-2015), suggesting that a substantial fraction of northward transport across the ridge is recirculated within the upper limb in the model.


As a typical water parcel flows cyclonically around the SPG, it forms dense NADW by cooling ($\Delta\theta_{SPG}$ = -4.0 ± 0.2 °C) and freshening ($\Delta S_{SPG}$ = -0.36 ± 0.03 g kg$^{-1}$) along-stream (Fig. 4). Figure 4b shows that, on average, the total light-



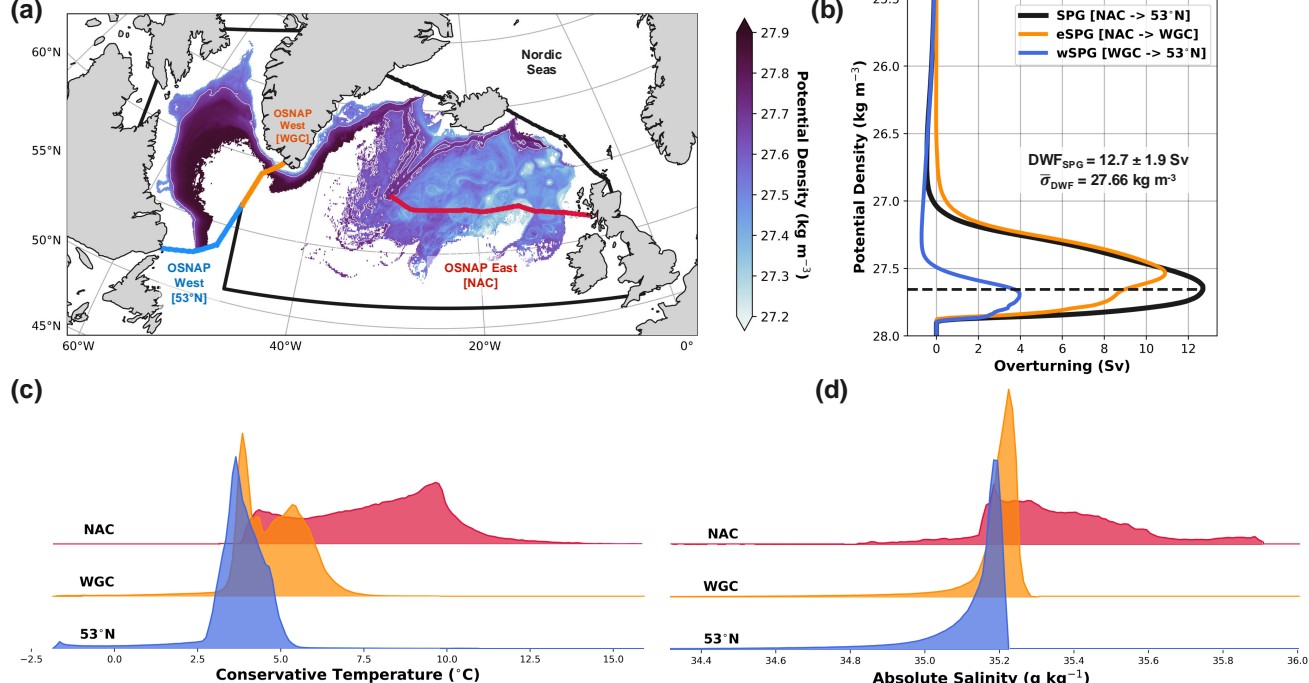

**Figure 4.** Water mass transformation along the path of the North Atlantic SPG. (a) Example evolution of potential density along the SPG pathway for the subset of water parcels which form NADW after flowing northward across OSNAP East in January 1995. The potential density sampled along each trajectory transiting from the NAC inflows across OSNAP East to 53°N along OSNAP West is binned in discrete latitude-longitude space ($\Delta x$, $\Delta y = 0.1°$) before calculating the average in each bin. The white contour superimposed shows the location of the 27.66 kg m$^{-3}$ isopycnal used to define the AMOC upper and lower limbs. The potential density values south of OSNAP East are due to water parcels recirculating south of the array en route to 53°N. (b) Time-mean (1975-2012) Lagrangian overturning stream functions in density-coordinates for the SPG pathway. The total diapycnal overturning along the full path of the SPG (black) is decomposed into the contributions of water mass transformation in the eastern (orange) and western SPG (blue). The black dashed line indicates the isopycnal of the maximum of the time-mean Lagrangian overturning stream function denoted as $\overline{\sigma}_{DWF}$. (c, d) Time-mean (1975-2012) volume transport distributions of water parcels along the path of the SPG in conservative temperature and absolute salinity coordinates. The three distributions shown in each panel correspond to the water parcel properties at their initial location along OSNAP East (NAC, red), their subsequent northward crossing of OSNAP West (WGC, orange) and their final southward crossing of OSNAP West (53°N, blue)

to-dense transformation of SPG water parcels peaks across the $\sigma_\theta = 27.66$ kg m$^{-3}$ isopycnal, which we herein refer to as
$\overline{\sigma}_{DWF}$. We hence define the along-stream Dense Water Formation (DWF) as the total volume flux of water parcels across this constant isopycnal between their initial release along OSNAP East and final southward crossing of OSNAP West (i.e., $\text{DWF}_{SPG}(t) = F(\overline{\sigma}_{DWF}, t)$). Notably, $\overline{\sigma}_{DWF}$ agrees closely with the isopycnal of maximum overturning recorded in OSNAP observations ($\overline{\sigma}_{MOC} = 27.63$ kg m$^{-3}$ during 2014-2020 in Fu et al. (2023)), although we acknowledge that this lies outside of





our study period. We herein refer to water parcels with a potential density less than or greater than $\bar{\sigma}_{DWF}$ as being found in

the upper or lower limb of the AMOC, respectively. Furthermore, we refer to lower limb water parcels collectively as NADW throughout the study since $\bar{\sigma}_{DWF}$ constitutes the time-mean upper isopycnal limit of NADW.

Of the $24.8 \pm 4.2$ Sv circulating around the SPG, Figure 4b indicates that $12.7 \pm 1.9$ Sv forms dense NADW (i.e., $\sigma_{53°N} >= \bar{\sigma}_{DWF}$). However, it is not possible for all of the water circulating around the SPG to form NADW along-stream, given that $5.6 \pm 1.4$ Sv flows northward across OSNAP East in the lower limb. Therefore, on average, we find that $12.7 \pm 1.9$ Sv of the 19.2

$\pm 3.0$ Sv flowing northward across OSNAP East in the upper limb forms NADW before returning southward across 53°N.

We additionally decompose the total DWF along-stream into the separate contributions made in the eastern and western SPG by calculating partial Lagrangian overturning stream functions using the properties of water parcels on their northward crossing of the OSNAP West array via the West Greenland Current (WGC; Fig. 4a). In agreement with OSNAP observations, we find that $\text{DWF}_{SPG}$ is dominated by cooling-driven diapycnal transformation in the eastern SPG ($9.0 \pm 1.7$ Sv; Fig. 4b).

In contrast, DWF is much weaker in the Labrador Sea ($3.7 \pm 0.9$ Sv) since there is greater density compensation between cooling and freshening along-stream (Fig. 4c-d). The mean strength of along-stream DWF in the Labrador Sea agrees well with the magnitude of diapycnal overturning observed along OSNAP West ($3.0 \pm 1.5$ Sv during 2014-2020 in Fu et al. (2023)). An equivalent comparison of DWF with observations in the eastern SPG is impeded by the contribution of the Nordic Seas overflows in the Eulerian diapycnal overturning stream function calculated at OSNAP East. However, we note that our finding

that the eastern SPG accounts for $9.0 \pm 1.7$ Sv of the total along-stream DWF agrees closely with both the results presented in Tooth et al. (2023a) and previous estimates from observations and ocean reanalyses, which suggest that between 9-10 Sv of diapycnal transformation takes place in the Iceland and Irminger Basins (e.g., Sarafanov et al., 2012; Chafik and Rossby, 2019; Koman et al., 2022; Buckley et al., 2023; Fu et al., 2024).

### 3.2 What governs dense water formation along the path of the subpolar gyre?

Figure 5a shows that the amount of dense NADW formed along the path of the SPG varies substantially across seasonal to decadal timescales. We next explore whether the initial properties of an upper limb water parcel on release across OSNAP East have any influence on the likelihood of forming dense NADW downstream.

Consistent with Tooth et al. (2023b), we find that variability in the composition of the upper limb at OSNAP East is dominated by seasonality (Fig. 5b); upper limb water parcels are, on average, lighter when crossing OSNAP East northwards

during autumn and denser when crossing during spring. However, Figure 5b shows that $\text{DWF}_{SPG}$ is not significantly correlated ($p > 0.05$) with the volume-weighted average potential density (nor the conservative temperature or absolute salinity) of water parcels flowing northward across OSNAP East in the upper limb. Furthermore, we find no statistically significant relationship between annual-means of $\text{DWF}_{SPG}$ and upper limb potential density (buoyancy), suggesting that along-stream DWF is not influenced by the arrival of upper limb buoyancy anomalies into the eastern SPG on either seasonal or interannual timescales.

In contrast, we find a strong positive correlation ($r = 0.86$, $p < 0.01$) between $\text{DWF}_{SPG}$ and the total northward upper limb transport flowing cyclonically from OSNAP East to OSNAP West (53°N), such that a larger volume transport of upper limb waters into the eastern SPG results in greater NADW formation downstream (Fig. 5c).





There are several important reasons why this enhanced DWF along the path of the SPG may not necessarily project onto Eulerian diapycnal overturning variability diagnosed along the full OSNAP array. Firstly, as we shall later see in Section 3d,

the formation of dense water occurs at many different locations around the SPG, meaning that there is a wide range of transit times for NADW to reach 53°N (Fig. 3a). Second, when calculating the Eulerian overturning along the full OSNAP array, the properties and volume fluxes of water parcels whose histories include dense water formation in the Nordic Seas and the Arctic Ocean are convolved with those transformed within SPG. (The thermohaline properties of water parcels enter this Eulerian overturning calculation because it involves integration within density classes.)

To better understand the relationship between upper limb volume transport and along-stream diapycnal transformation, we can express the total DWF along the path of the SPG as:

$$DWF_{SPG}(t) = \kappa(t)V_{SPG[UL]}(t) \tag{5}$$

where $V_{SPG[UL]}(t)$ represents the SPG upper limb volume transport arriving at OSNAP East at a time $t$ and $\kappa(t)$ represents the fraction of this upper limb volume transport that will form dense NADW downstream. Furthermore, by decomposing each term into its steady and fluctuating components (i.e., $\kappa(t) = \bar{\kappa} + \kappa'(t)$ and $V_{SPG[UL]}(t) = \bar{V}_{SPG[UL]} + V'_{SPG[UL]}(t)$), we can clearly

see that variations in $DWF_{SPG}$ are potentially due to a complex combination of changes in the amount of upper limb water flowing northward across OSNAP East $V'_{SPG[UL]}(t)$ and changes in the efficiency by which water parcels are transferred from the upper to the lower limb along-stream $\kappa'(t)$:

$$DWF_{SPG}(t) = \bar{\kappa}\bar{V}_{SPG[UL]} + \bar{\kappa}V'_{SPG[UL]}(t) + \kappa'(t)\bar{V}_{SPG[UL]} + \kappa'(t)V'_{SPG[UL]}(t) \tag{6}$$

Surprisingly, Figure 5c suggests that the efficiency of along-stream diapycnal transformation $\kappa'(t)$ is not the rate-limiting factor governing DWF along the path of the SPG. Instead, variations in $DWF_{SPG}$ are proportional to the amount of upper limb water imported into the eastern SPG via the branches of the NAC (i.e., $DWF_{SPG}(t) \propto V'_{SPG[UL]}(t)$). This implies that along-stream diapycnal transformation is sufficient to transfer a steady fraction $\bar{\kappa}$ of upper limb water parcels into the lower limb, irrespective of their initial thermohaline properties on their northward crossing of OSNAP East. Furthermore, the SPG

upper limb volume transport is also strongly correlated with the total volume transport along the path of the SPG ($r = 0.97$, $p < 0.01$), indicating that $DWF_{SPG}$ is closely related to the overall strength of the SPG circulation.

### 3.3 How predictable is subpolar gyre dense water formation?

Since we have shown that DWF along the path of the SPG is proportional to the upper limb volume transport imported into the eastern SPNA, we next develop a simple linear model to predict the amount of dense NADW formed along-stream. By

assuming that the efficiency of water mass transformation from the upper to the lower limb is time-independent (i.e., $\kappa = \bar{\kappa}$), we can formulate a linear model for $DWF_{SPG}$ as follows:

$$DWF_{SPG}(t) = \bar{\kappa}V_{SPG[UL]}(t) + \epsilon(t) \tag{7}$$



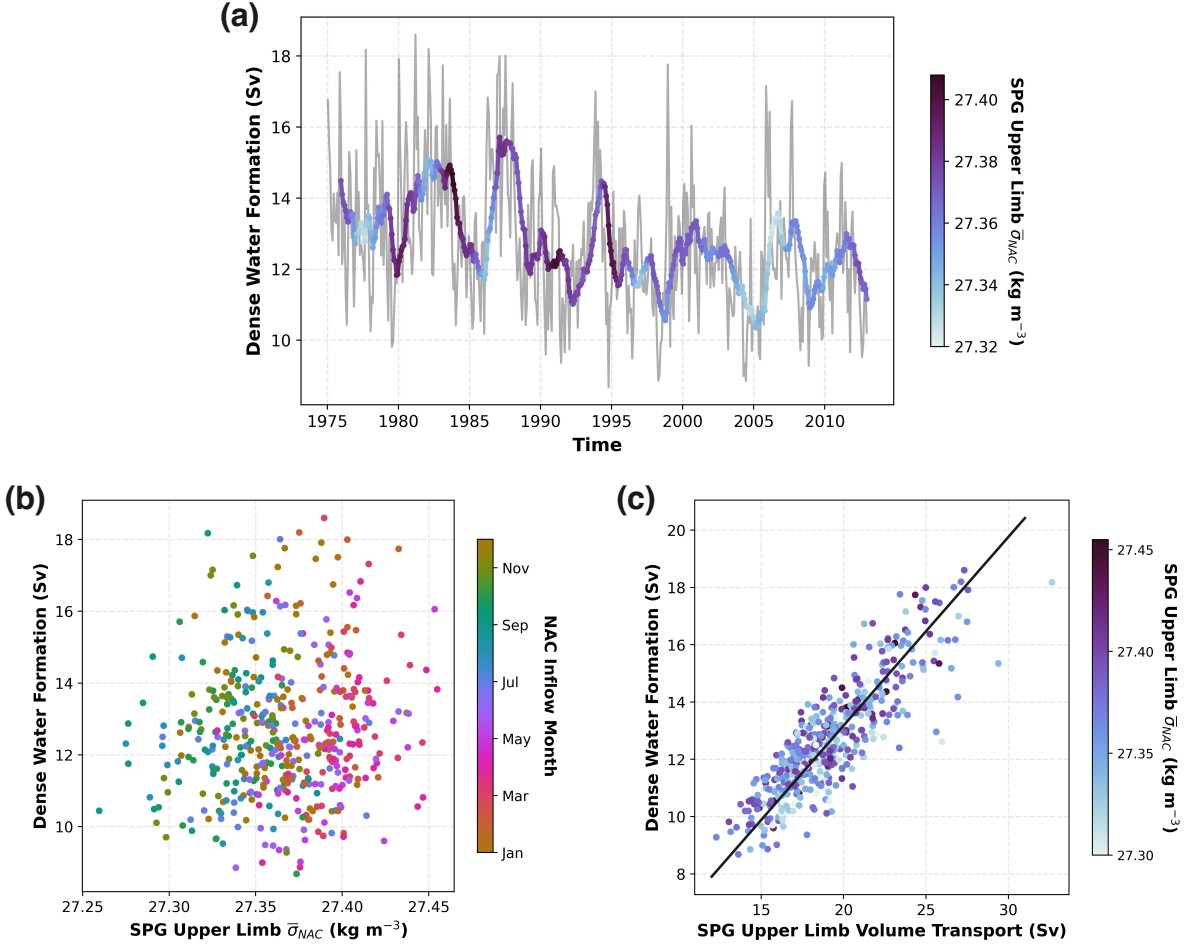

**Figure 5.** Controls on dense water formation along the path of the North Atlantic SPG. (a) Dense water formation (i.e., total volume flux of water parcels across $\overline{\sigma}_{DWF}$) along the path of the SPG (NAC to 53°N) plotted according to the common inflow time when water parcels flow northward across OSNAP East. The bold line is obtained by applying a 12-month running mean filter to the monthly dense water formation (light grey line) coloured by the volume-weighted mean potential density of the SPG upper limb (i.e., $\sigma_{NAC} \leq \overline{\sigma}_{DWF}$) water parcels. (b) Monthly dense water formation plotted against the volume-weighted mean potential density of SPG upper limb water parcels coloured by their month of release across OSNAP East in the NAC. (c) Monthly dense water formation plotted against the total volume transport of SPG upper limb water parcels coloured by their volume-weighted mean potential density on release across OSNAP East.

where $\bar{\kappa}$ is the constant fraction of the upper limb volume transport which forms NADW along-stream and $\epsilon(t)$ represents the residual error, which is given by:

$\quad \epsilon(t) = \kappa'(t)\bar{V}_{SPG[UL]} + \kappa'(t)V'_{SPG[UL]}(t)$ (8)





We find that $\bar{\kappa} = 0.66$, implying that, on average, 66% of upper limb waters flowing northward across OSNAP East are transferred into the lower limb prior to their southward crossing of OSNAP West (53°N). Figure 6a shows the strong predictive skill of the linear model on both monthly (RMSE = 1.1 Sv) and interannual (RMSE = 0.8 Sv) timescales, accounting for 74% and 68% of the variance in monthly and interannual (12-month running mean filtered) DWF, respectively.

To better understand the sources of error in our linear model, we decompose the residual DWF $\epsilon(t)$ into its two components in Figure 6b. We find that the residual DWF is almost exclusively explained by fluctuations in the efficiency of along-stream diapycnal transformation from the upper to the lower limb acting on the time-mean volume transport of the SPG upper limb ($\kappa'(t)\bar{V}_{SPG[UL]}$). More specifically, the linear model overestimates the amount of dense NADW formed during the late 1970s and mid-1980s, indicating that the time-independent efficiency of diapycnal transformation of the linear model is too large

during these periods (i.e., $\bar{\kappa} > \kappa(t)$). In contrast, during the early and late 2000s, the efficiency of transformation is slightly underestimated (i.e., $\bar{\kappa} < \kappa(t)$) resulting in an underestimation of the DWF downstream. Despite these differences, the high predictive skill of the linear model, especially on interannual to decadal timescales, suggests that changes in diapycnal transformation efficiency play a secondary role in governing variations in $DWF_{SPG}$ when compared to variability in the upper limb transport imported into the eastern SPG.

Given that $\bar{\kappa} = 66\%$ of the upper limb waters arriving across OSNAP form dense NADW along the path of the SPG, we next consider what happens to the remaining $(1 - \bar{\kappa}) = 34\%$ of upper limb waters which do not form dense NADW before crossing OSNAP West at 53°N. Of the $6.4 \pm 1.7$ Sv of upper limb water which does not form dense NADW along the path of the SPG, we find that $1.5 \pm 0.5$ Sv becomes lighter through entrainment into the fresh Labrador Coastal Current. Meanwhile, the majority of outstanding upper limb transport ($5.0 \pm 1.4$ Sv) becomes denser but not dense enough to be transferred into the

lower limb on crossing OSNAP West via the Labrador Current. To determine the fate of these denser water parcels remaining in the upper limb at 53°N, we extend our original Lagrangian experiment by continuing to track upper limb water parcels after they cross the OSNAP West section. We find that almost all of the denser upper limb water parcels (94%) are either transformed into dense NADW south of the OSNAP West section or return to OSNAP East via the NAC to continue circulating around the SPG. Thus, of the total upper limb transport imported into the eastern SPG, we would expect that, on average, almost

92.5% will form dense NADW during one or more additional circuits of the SPG, whereas 7.5% will join the fresh, estuarine circulation confined to the shelves of the SPNA.

### 3.4   Two dense water formation pathways around the subpolar gyre

We have demonstrated that the total DWF along the path of the SPG can be skilfully predicted with knowledge of only the northward upper limb transport which flows cyclonically around the SPG and a time-independent parameter $\bar{\kappa}$, representing

the efficiency of diapycnal transformation along-stream.

Figure 7 reveals that there are, in-fact, two distinct pathways by which dense water is formed along the path of the SPG. By decomposing $DWF_{SPG}$ into the separate contributions made by the NAC branches flowing northward in the Iceland Basin and in the Rockall Trough (Figure 7a), we find that upper limb water parcels sourced from the Iceland Basin ($DWF_{IB}$) account for almost three quarters ($9.3 \pm 1.8$ Sv) of the time-mean $DWF_{SPG}$ and 88% of its variance on interannual timescales (Fig.



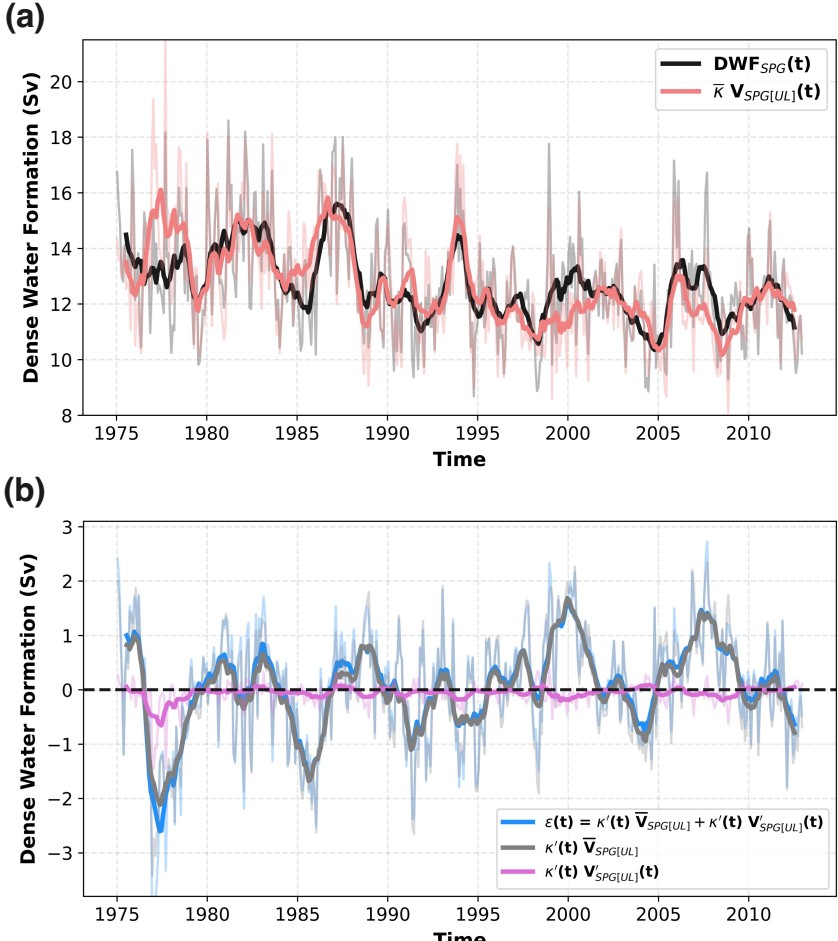

**Figure 6.** A linear model for Dense Water Formation (DWF) along the path of the North Atlantic SPG. (a) Monthly DWF along the path of the SPG (NAC to 53°N, black) and estimated DWF using a simple linear model $\bar{\kappa}V_{SPG[UL]}(t)$ (pink). (b) The residual DWF $\epsilon(t)$ which is not included in the simple linear model is decomposed into the contributions made by the fluctuations in the efficiency of diapycnal transformation from the upper to the lower limb acting on the time-mean transport of the upper limb and a non-linear term representing the correlation between fluctuations in the efficiency of diapycnal transformation and in the upper limb volume transport. The bold lines are obtained by applying a 12-month running mean filter to the monthly values.

7b). This contrasts with upper limb water parcels flowing northward in the Rockall Trough, which account for $3.4 \pm 0.7$ Sv ($\mathrm{DWF}_{RT}$) of along-stream DWF and around 31% of the interannual variability in $\mathrm{DWF}_{SPG}$. These differences in along-stream DWF are partly explained by the larger northward transport entering the Iceland Basin across OSNAP East ($21.2 \pm 4.0$ Sv) compared with the Rockall Trough ($3.9 \pm 0.8$ Sv in Fig. 3b). However, this is far from the complete picture, given that these





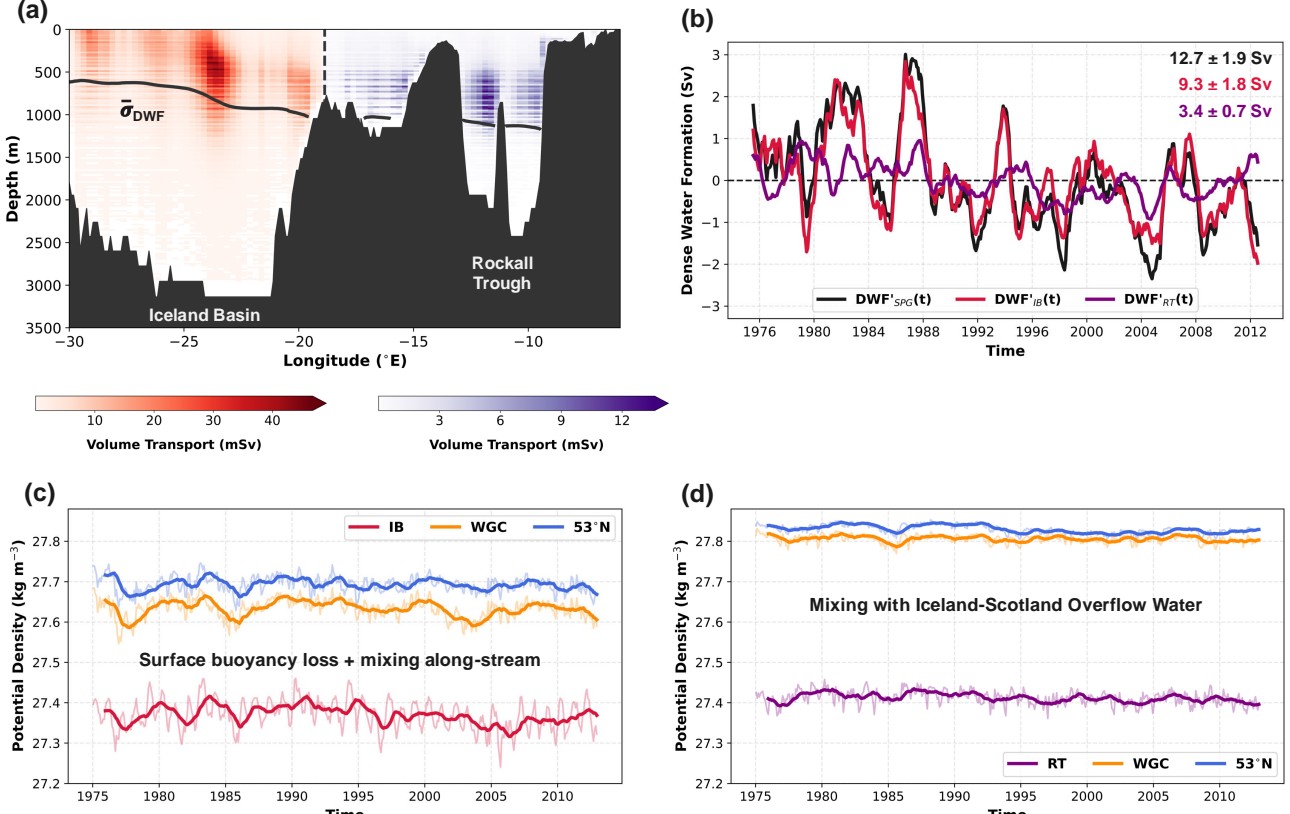

**Figure 7.** Dense water formation along the two circulation pathways of the North Atlantic SPG. (a) Time-mean (1975-2012) volume transport of SPG water parcels originating from the Iceland Basin (**IB**, red) and the Rockall Trough (**RT**, purple) across OSNAP East. Water parcel volume transports are binned in discrete longitude-depth space ($\Delta x = 0.25°$E, $\Delta z = 20$ m) according to their initial release locations along OSNAP East. The black solid line shows the time-mean (1975-2012) position of the $\bar{\sigma}_{DWF}$ isopycnal used to define along-stream dense water formation and hence delimits the upper and lower limbs of the AMOC in this study. (b) Anomalies in the total DWF along the path of the SPG (DWF'$_{SPG}$, black) decomposed into the contributions made by upper limb water parcels sourced from the Iceland Basin (DWF'$_{IB}$, red) and the Rockall Trough (DWF'$_{RT}$, purple). We apply a 12-month running mean filter to the monthly DWF anomalies in order to highlight interannual to decadal variability. (c,d) Volume-weighted mean potential density evolution of upper limb water parcels, sourced from the **IB** (c) and the **RT** (d), which experience a net positive diapycnal transformation along-stream ($\Delta\sigma > 0$ kg m$^{-3}$). The bold lines are obtained by applying a 12-month running mean filter to the monthly mean potential density values recorded on initialisation across OSNAP East (**IB / RT**) and on crossing OSNAP West via both the West Greenland Current (**WGC**) and the Labrador Current (53°N). Note, values are plotted according to the time when water parcels flow northward across the OSNAP East section in panels (b-d).





two dense water pathways also exhibit distinct modes of diapycnal transformation downstream (Figures 7c-d), resulting in
markedly different efficiencies in the transformation of water parcels from the upper to the lower limb.

We find that 59% ($\bar{\kappa}_{IB} = 0.59$, Fig. 8a) of upper limb waters entering the Iceland Basin via the northern and central branches
of the NAC form upper NADW downstream ($\bar{\sigma}_{DWF} < \sigma_{53°N} < 27.80$ kg m$^{-3}$). Consistent with the dominant Lagrangian
overturning pathway identified in Tooth et al. (2023a), we find that upper limb water parcels undergo progressive transformation
by surface buoyancy loss and diapycnal mixing along-stream (Fig. 4a and 7c). In contrast, practically all ($\bar{\kappa}_{RT} = 0.97$, Fig.
8a) of the upper limb waters arriving subsurface in the Rockall Trough form dense lower NADW ($\sigma_{53N} > 27.80$ kg m$^{-3}$) by
intense, localised diapycnal mixing with Iceland-Scotland Overflow Waters (ISOW) at the exit of the Faroe-Bank Channel
(Fig. 4a and 7d). This finding is consistent with previous studies (e.g., Sarafanov et al., 2012; Devana et al., 2021; Chafik and
Holliday, 2022), which identify a 'short-cut' pathway for subtropical-origin waters to penetrate the deep ocean on sub-decadal
timescales by mixing with overflow waters south of the Iceland-Faroes Ridge. The clear distinction between the character of
dense water formed along these two pathways is evident from the two peaks in the conservative temperature distribution of
SPG water parcels on their northward crossing of OSNAP West via the WGC in Figure 4c.

Concordant with our earlier analysis in Section 3b, Figures 7c-d show that potential density (buoyancy) anomalies conveyed
by upper limb water parcels arriving in both the Iceland Basin and the Rockall Trough are strongly damped downstream. This
is particularly evident in the Iceland Basin (Fig. 7c), where we find that variations in the average potential density of upper limb
water parcels can only explain around 30% of their downstream variability at 53°N (i.e., $r(\sigma_{IB}, \sigma_{53°N}) = 0.56$, $p < 0.01$). The
increasing homogeneity of the two dense water pathways on reaching the exit of the Labrador Sea (53°N) is also evident in
Figures 8b-c, which present the relationship between the initial potential density of upper limb water parcels (excluding those
entrained into the fresh estuarine circulation) and their net change in potential density along-stream. We find that, on seasonal
to interannual timescales, upper limb waters sourced from both the Iceland Basin and the Rockall Trough experience a net
densification which is inversely proportional to their potential density on flowing northwards across OSNAP East (Figs. 8b-c).

## 3.5 What drives decadal variability in subpolar gyre dense water formation?

We have seen that the amount of dense water formed along the path of the SPG exhibits substantial variations on decadal
timescales (Fig. 5a), which principally result from changes in the transport of upper limb water arriving across OSNAP East
via the branches of the NAC. More specifically, we find that DWF$_{SPG}$ transitions from a relatively strong period between
1975-1987 (13.7 ± 2.0 Sv) to a weaker, less variable period extending from 2000-2012 (12.1 ± 1.5 Sv; see Fig. 7b). Since
the amount of upper limb water flowing northward in the NAC is also strongly correlated with the total volume transport
circulating around the SPG, we next investigate the mechanisms responsible for generating variability in the strength of the
SPG and, hence, DWF on decadal to multi-decadal timescales.

Previous numerical modelling studies have highlighted the important role of localised surface buoyancy forcing, driven by
low-frequency changes in the NAO, in modulating decadal variability in the subpolar ocean circulation (Jackson et al., 2022;
Yeager and Danabasoglu, 2014; Yeager et al., 2015; Yeager, 2020; Böning et al., 2006; Robson et al., 2012; Delworth and
Zeng, 2016; Kim et al., 2018; Khatri et al., 2022). Consistent with the mechanisms proposed in these studies, we find that the






**Figure 8.** Variability in dense water formation along the two circulation pathways of the North Atlantic SPG. (a) Dense water formation and the total volume transport of SPG upper limb water parcels sourced from the Iceland Basin (**IB**, red) and the Rockall Trough (**RT**, purple) inflows across OSNAP East. The values of $\bar{\kappa}$ overlaid represent the fraction of upper limb waters sourced from the **IB** and **RT** which form dense NADW before reaching 53°N at OSNAP West. (c, d) Volume-weighted mean potential density of SPG upper limb water parcels ($\bar{\sigma}_{IB}$, $\bar{\sigma}_{RT}$) and their volume-weighted mean net densification along-stream ($\bar{\sigma}_{53°N}$ - $\bar{\sigma}_{IB}$, $\bar{\sigma}_{53°N}$ - $\bar{\sigma}_{RT}$) coloured by the year of their initial northward crossing of OSNAP East. We include only the SPG upper limb water parcels experiencing a net positive diapycnal transformation (i.e, $\Delta\sigma > 0$ kg m$^{-3}$) along-stream to exclude those entrained into the fresh estuarine circulation in the Labrador Sea.

generation of subsurface density anomalies and the densification of the AMOC lower limb are both important precursors to sustained positive anomalies in DWF along the path of the SPG. In particular, Figures 9a-b show that persistent positive phases of the NAO during the mid-1980s and early 1990s were responsible for enhanced surface heat loss and, therefore, an intensi-

fication in deep convection in the SPG interior. This resulted in anomalously strong surface-forced water mass transformation





**Figure 9.** Mechanisms governing decadal variability in dense water formation along the path of the North Atlantic SPG. (a) Winter [DJFM] station-based North Atlantic Oscillation (NAO) index. (b) Surface forced Water Mass Transformation (WMT) anomalies calculated over the Lagrangian experiment domain (see Fig. 1a) in $\sigma_2$ potential density coordinates (referenced to 2000 m). (c) Layer thickness anomalies of Labrador Sea Water ($\Delta z_{LSW}$) in the basin interior (where bathymetry exceeds 2000 m) along the OSNAP West (blue) and OSNAP East (orange) arrays. (d) Basin interior (where bathymetry exceeds 2000 m) Sea Surface Height (SSH) anomaly relative to the section-wide mean SSH along the OSNAP West (blue) and OSNAP East (orange) arrays. (e) Variations in the total volume transport of SPG upper limb water parcels ($V_{SPG[UL]}$, black) decomposed into contributions from water parcels sourced from the Iceland Basin ($V_{IB[UL]}$, red) and the Rockall Trough ($V_{RT[UL]}$, purple). (f) Dense water formation along the path of the SPG (black) and the volume-weighted mean transit time of upper limb water parcels circulating around the SPG (teal). Note that values in panels (e-f) are plotted according to the time when water parcels flow northward across the OSNAP East section. All anomalies are determined by removing the long-term time-mean (1975-2012) from monthly values before applying a 36-month running mean filter.





in the LSW density range (i.e., $\sigma_2 > 37.0$ kg m$^{-3}$ in Fig. 9b), which increased the thickness of the LSW layer in the central Labrador and western Irminger Seas (Fig. 9c). The densification of the SPG interior also manifests at the surface through a depression in the sea-surface height (SSH) field (see Fig. 9d), which induces a delayed spin-up of the SPG circulation by

steepening the SSH gradient across the basin (Häkkinen and Rhines, 2004; Eden and Willebrand, 2001; Kostov et al., 2023). In agreement with recent studies (Chafik et al., 2022; Roussenov et al., 2022; Mercier et al., 2024), we find that SSH (density) anomalies in the Irminger Sea interior play an important role in determining the northward geostrophic transport of the upper limb by modulating the pressure gradient across the NAC. Specifically, Figure 9e shows that the stronger upper limb transport arriving in the Iceland Basin between 1975-1987 is associated with a period of anomalously low sea-surface heights in the

Irminger Sea interior, whereas elevated sea-surface heights are concomitant with the weaker upper limb transport recorded during 2000-2012 (Fig. 9d).

To further demonstrate that multi-decadal variations in the upper limb transport entering the Iceland Basin are concordant with low-frequency changes in SPG dynamics, Figure 9f shows the average transit times taken for upper limb water parcels to circulate around the SPG. We can clearly see that the greater upper limb volume transport across OSNAP East during 1975-

1987 coincides with a faster SPG circulation. Meanwhile, during 2000-2012, the slower SPG circulation is responsible for the weaker upper limb transport arriving in the NAC. Finally, since we have already shown that DWF along the path of the SPG depends linearly on the upper limb transport flowing northward across OSNAP East, Figure 9f shows that multi-decadal changes in SPG DWF are largely determined by the response of the gyre circulation to remote (i.e., Labrador Sea) surface buoyancy forcing. The strong interannual variability superimposed on this multi-decadal variability in DWF likely reflects the

faster wind-driven response of the SPG circulation to changes in the NAO (e.g., Eden and Willebrand, 2001; Khatri et al., 2022). For example, Wang et al. (2021) show that wind-stress curl induced variations in the transport of the NAC branches arriving in the Iceland Basin and the Rockall Trough play an important role in driving interannual variability in the upper limb transport across OSNAP East.

In summary, we have shown that decadal surface buoyancy forcing anomalies in the central Labrador and Irminger Seas can

remotely influence NADW formation taking place along the path of the SPG by modulating the strength of the SPG circulation and hence the availability of upper limb waters.

## 4   Discussion

Despite substantial multi-decadal variability in the water mass properties of the subpolar North Atlantic Ocean, the extent to which large-scale thermohaline changes impact the formation of NADW in the subpolar gyre remains poorly understood. Here,

we have investigated the physical mechanisms governing DWF along the path of the SPG by taking a Lagrangian perspective to determining how much dense water is formed as water parcels circulate around the SPG in an eddy-rich ocean model. Our analysis has revealed three important insights into the nature of dense water formation along the boundary current of the SPG: (a) the coupling between the subpolar gyre and overturning circulations, (b) the decoupling between upper limb thermohaline





anomalies and dense water formation, and (c) the influence of remote surface buoyancy forcing on decadal subpolar overturning
variability.

## 4.1   The coupling between the subpolar gyre & overturning circulation components

In Section 3c, we demonstrated that DWF along the path of the SPG can be skilfully predicted using a simple linear model
in which a constant fraction $\bar{\kappa} = 66\%$ of the available upper limb volume transport is transformed into NADW during each
circuit of the SPG. We have seen that one interpretation of $\bar{\kappa}$ is a measure of the efficiency of diapycnal transformation from the
upper to the lower limb. However, this can also be conceptualised as a measure of the relative alignment between the gyre and
diapycnal overturning circulations at subpolar latitudes. To illustrate this, we can consider the idealised case in which the SPG
and overturning circulations are perfectly aligned (i.e., $\bar{\kappa} = 100\%$), such that all of the upper limb waters flowing northward
in the NAC are transformed into lower limb waters on returning southward in the Labrador Current. In this case, the volume
transport flowing around the SPG would be equivalent to the DWF along-stream and we could quantify the subpolar overturning
by simply measuring the strength of the SPG circulation via the total volume transport advected in the branches of the NAC.
However, we know from observations that $\bar{\kappa} < 100\%$ since the isopycnal delimiting upper NADW does not persistently outcrop
one the western side of the Labrador Sea (see Fig. 4 in Zantopp et al., 2017), meaning that the Labrador Current transports
both upper and lower limb waters southward. Thus, in reality, the SPG circulation projects onto a diapycnal overturning cell
(and thus the formation of NADW) with a time-evolving efficiency characterised by $\kappa(t)$.
Since dense water is also formed via progressive diapycnal transformation along the boundary current encircling the Nordic
Seas (e.g., Eldevik et al., 2009; Isachsen et al., 2007; Mauritzen, 1996), it would be interesting to extend the Lagrangian
methodology introduced in this study to establish if a similar linear relationship can be found between the northward transport
of Atlantic Waters across the Greenland-Scotland Ridge and the along-stream formation of lower NADW (i.e., $\mathrm{DWF}_{NS} =$
$\bar{\kappa}_{NS} V_{AW}$). Previous studies have estimated that approximately 70-75% of the Atlantic Water inflow to the Nordic Seas partic-
ipates in the thermohaline circulation to form dense overflow waters (Østerhus et al., 2019; Le Bras et al., 2021), suggesting
that $\bar{\kappa}_{NS} \approx 0.7 - 0.75$.

## 4.2   Decoupling between upper limb thermohaline anomalies & dense water formation

We have also seen that the likelihood of downstream transformation into the lower limb is strongly dependent on where
upper limb waters arrive into the eastern SPG. In particular, we showed that upper limb waters arriving subsurface in the
Rockall Trough are almost guaranteed ($\bar{\kappa}_{RT} = 97\%$) to form dense lower NADW via vigorous mixing with ISOW, whereas
only $\bar{\kappa}_{IB} = 59\%$ of upper limb waters arriving in the Iceland Basin will form upper NADW downstream. Consistent with
the conclusion of Fu et al. (2020), we found that the amount of dense water formed along each pathway is independent of the
initial properties of water parcels arriving in the NAC on seasonal to interannual timescales, indicating that upper limb potential
density anomalies do not feed back onto the strength of DWF and hence diapycnal overturning.
To better understand this decoupling between the strength of diapycnal overturning and upper ocean properties in the SPNA,
we consider the length scales on which upper limb thermohaline anomalies evolve along their path from the Iceland Basin to





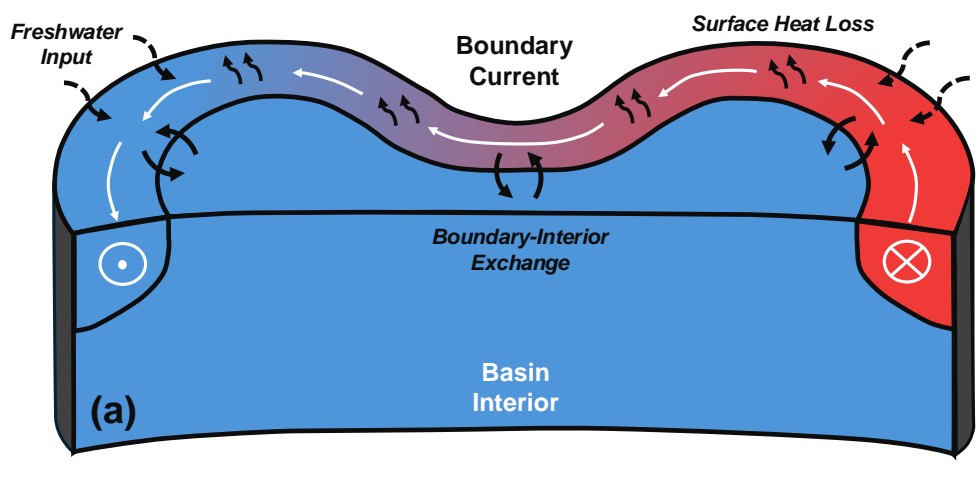

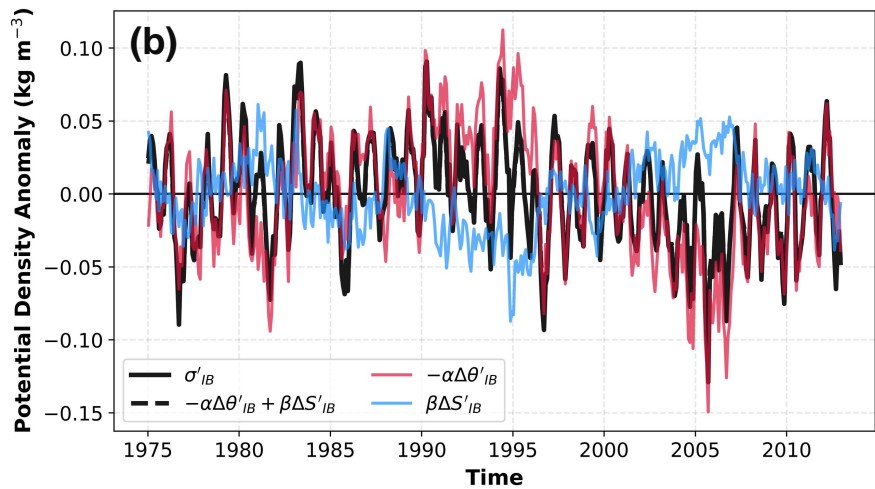

**Figure 10.** (a) Schematic of an idealised boundary current circulating cylonically around a subpolar basin. (b) Decomposition of the volume-weighted mean potential density anomaly of upper limb waters flowing northward across OSNAP East in the Iceland Basin. Potential density anomalies from the time-mean ($\sigma'_{IB}$, black) are decomposed into their respective diathermal ($-\alpha\Delta\theta'_{IB}$, red) and diahaline ($\beta\Delta S'_{IB}$, blue) components using a linearised equation of state for seawater. Note that the potential density anomaly (solid black line) of upper limb waters is entirely reconstructed by the sum of diathermal ($-\alpha\Delta\theta'_{IB}$) and diahaline ($\beta\Delta S'_{IB}$) components (underlying dashed black line).





the western subpolar North Atlantic. For an idealised boundary current which exchanges buoyancy with both the overlying atmosphere and the basin interior (Fig. 10a), temperature and salinity adjust exponentially along-stream towards equilibrium values (Wåhlin and Johnson, 2009). The cyclonic boundary current will therefore reach an equilibrium density provided that

its length exceeds the adjustment length scales with respect to both temperature and salinity. Following Wåhlin and Johnson (2009), and assuming the inflow to the Iceland Basin is 200 km wide, a temperature relaxation coefficient of 80 W m$^{-2}$ and an exchange rate between the boundary current and basin interior of $M = 2.0$ m$^2$ s$^{-1}$, we estimate the adjustment length scales for the upper limb waters sourced from the Iceland Basin to be approximately 2400 km and 7100 km for temperature and salinity, respectively. (Note that in the case of salinity this estimate is an upper limit, because we have assumed that the along-stream

addition of freshwater $F$ is small compared with the freshwater exchanged with the basin interior.) Comparing these length scales with the typical path length of upper limb water parcel trajectories travelling from the Iceland Basin to OSNAP West at 53°N (approximately 6000 km) we expect that the boundary current is fully adjusted in terms of temperature, but not in terms of salinity.

The temperature of water reaching 53°N is therefore virtually independent of both the volume transport and initial tem-

perature of upper limb waters flowing north across OSNAP East, whereas some signature of salinity anomalies arriving in the Iceland Basin may persist. Figure 10b shows that upper limb potential density anomalies arriving in the Iceland Basin are dominated by temperature rather than salinity fluctuations on monthly to decadal timescales (i.e., $r(-\alpha\theta'_{IB}, \sigma'_{IB}) = 0.82$, $p < 0.01$). This dominance of thermal anomalies in the upper limb, combined with their efficient damping by air-sea fluxes and mixing during their transit of the gyre, explains both the strong decoupling between decadal changes in upper-ocean properties

and SPG DWF (e.g., Fu et al., 2020) and the narrow potential density range of upper NADW in observations (27.68 - 27.74 kg m$^{-3}$; Rhein et al., 2011; Kieke et al., 2007).

The use of monthly-mean model velocity and tracer fields to evaluate Lagrangian water parcel trajectories is an important limitation of this study, albeit necessary to make our calculations tractable. This is because we likely underestimate the dispersive nature of Lagrangian trajectories and, hence, the volume exchanges between the boundary current and the interior of the

Labrador and Irminger Seas (Georgiou et al., 2020, 2021). By using daily or 5-day mean velocity and tracer fields, we would expect shorter circulation times (Blanke et al., 2012) and greater boundary-interior exchanges along the path of the SPG (Roach and Speer, 2019). However, the effects of eddy exchange between the boundary current and the basin interior are implicitly captured in the tracer fields, sampled along water parcel trajectories (Chenillat et al., 2015), and are therefore included in our estimates of along-stream DWF.

### 4.3 The influence of remote surface buoyancy forcing on decadal subpolar overturning variability

A further implication of the strong decoupling between the properties of upper limb waters arriving in the eastern SPG and the strength of DWF downstream is that potential density anomalies advected along the path of the SPG do not play an active role in driving subpolar overturning variability on seasonal to decadal timescales, since they are unable to persist downstream. This supports the conclusion of Buckley et al. (2012) that, although decadal AMOC variability can generate upper-ocean ther-

mohaline anomalies, these anomalies are not themselves responsible for generating decadal subpolar overturning variability.



Instead, decadal variations in the DWF along the path of the SPG are driven remotely by surface buoyancy forcing localised in the central Labrador and Irminger Seas.

Concordant with recent studies (e.g., Roussenov et al., 2022; Kostov et al., 2023, 2024), we find that enhanced surface buoyancy loss during persistent positive phases of the NAO drives a geostrophic increase in the northward upper limb transport into the Iceland Basin, which is consistent with an intensification of the cyclonic SPG circulation in response to the densification of the Irminger Sea interior. Since along-stream DWF is proportional to the amount of upper limb water flowing northward across OSNAP East, we might anticipate that this spin-up of the SPG circulation would directly translate into an increase in the strength of the basin-scale diapycnal overturning circulation. However, in order for DWF to imprint onto the Eulerian overturning at lower latitudes, lower limb waters must also be exported out of the SPG and into the subtropical North Atlantic (Buckley et al., 2023; Zou and Lozier, 2016). By comparing the long-term mean subpolar AMOC strength (∼16 Sv) to the typical southward transport of NADW at 53°N (∼30 Sv; Fröhle et al., 2022; Zantopp et al., 2017), Buckley et al. (2023) estimate that only around half of all NADW is exported to the subtropics, whilst the remainder recirculates in the SPG. Although it is beyond the scope of this study, it would be interesting to investigate whether decadal changes in DWF can influence the rate at which NADW is exported to the subtropical North Atlantic and, hence, act to modulate the advective propagation of overturning anomalies downstream.

## 5 Conclusions

In this study, we have identified the controls on the formation of NADW along the boundary current of the SPG by tracing the evolution of upper limb waters from their arrival in the eastern SPNA to their southward return along the western boundary of the Labrador Sea. We have shown that neither along-stream diapycnal transformation, nor the arrival of thermohaline anomalies in the NAC, are the rate limiting factors governing DWF. Instead, the amount of dense water formed downstream can be skilfully predicted based solely on the volume transport of upper limb waters circulating cyclonically around the SPG, which is modulated by remote surface buoyancy forcing in the interior Labrador and Irminger Seas on decadal timescales. This central finding suggests that low-frequency changes in subpolar overturning must also manifest in the SPG circulation, thereby underscoring the importance of monitoring the state of the SPG for both decadal and longer-term climate predictions as previously highlighted by Bingham et al. (2007) and Buckley et al. (2012).

More broadly, our findings imply that the projected decline of the AMOC over the 21st century will be closely related to the evolution of the SPG circulation and its representation in coupled climate models (Hirschi et al., 2020). On the one hand, the robust weakening and contraction of the SPG circulation in response to external anthropogenic forcing in coupled climate models (Sgubin et al., 2017; Swingedouw et al., 2021, 2020; Born and Stocker, 2014) is entirely consistent with a decline in along-stream NADW formation. However, this is undermined by the substantial biases exhibited by current generation coupled climate models, which favour NADW formation through excessive deep convection in the basin interior (Heuzé, 2017, 2021) rather than via continuous diapycnal transformation along the boundary current of the SPG (Hirschi et al., 2020). Moving forward, diagnosing precursor quantities of low-frequency subpolar dense water formation, such as Labrador Sea subsurface



density (Ortega et al., 2017, 2021) or SPG transport indices (Curry and McCartney, 2001; Koul et al., 2020), may prove to be
an effective means of quantifying future AMOC weakening alongside monitoring the state of the circulation itself.

*Code and data availability.* Monthly mean outputs from the ORCA0083-GO8p7 ocean sea-ice hindcast (Megann et al., 2022) are available
from https://dx.doi.org/10.5285/399b0f762a004657a411a9ea7203493a. Overturning in the Subpolar North Atlantic Program (OSNAP) data
were collected and made freely available at https://doi.org/10.35090/gatech/70342 by the OSNAP project and all the national programs that
contribute to it (www.o-snap.org). The Lagrangian trajectory code TRACMASS was developed by Aldama-Campino et al. (2020) and is
available from https://github.com/oj-tooth/Tracmass_v7.1. The Lagrangian trajectory crossings of the OSNAP arrays can be obtained from
https://doi.org/10.5281/zenodo.14870254. The analysis of Lagrangian trajectories was performed using the Lagrangian Trajectories Toolbox,
an open-source Python library available from https://github.com/oj-tooth/lt_toolbox.

*Author contributions.* OJT, HLJ & CW conceptualised the study. OJT designed and performed the Lagrangian experiments and developed
the LT-Toolbox analysis software. OJT prepared the original manuscript draft. OJT, HLJ & CW contributed to the interpretation of results,
reviewed and edited the manuscript.

*Competing interests.* The authors declare that they have no conflict of interest.

*Acknowledgements.* O.J. Tooth is grateful for the financial support of the UK Natural Environment Research Council (NE/S007474/1) and
the Atlantic Climate and Environment Strategic Science (AtlantiS) grant (NE/Y005589/1). H.L. Johnson was supported by the NERC-
NSF SNAP-DRAGON project (NE/T013494/1). C.W. was supported by the CANARI project (NE/W004984/1). We would like to thank
Alex Megann for performing the ORCA0083-GO8p7 simulation as part of the North Atlantic Climate System Integrated Study (ACSIS)
programme. We are also grateful to Laura Jackson, who kindly provided the original code to extract the coordinates of OSNAP arrays from
the NEMO model grid.



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
