# Peer review of "Controls on dense water formation along the path of the North Atlantic subpolar gyre"

_EGUsphere, 2025_

## Referee Comment (RC1)

**Summary.** Lagrangian analysis of how the northward extension of the North Atlantic Current (across the Eastern half of the OSNAP East section) contributes to the formation of North Atlantic Deep Waters by exposing light "upper limb" waters to buoyancy loss due to surface buoyancy fluxes or turbulent entrainment into denser water masses. Variability in the (Lagrangian) North Atlantic Deep Water formation rate is attributed to the overall strength of the Subpolar Gyre, which is connected to the water mass properties of the gyre's interior (via an assumed thermal wind relationship). These results are contrasted with the idea that the advection of buoyancy anomalies controls the formation of deep waters, as in the salt-advection feedback thought to control the overall stability of the AMOC.

**Overall Assessment.** This is an excellent manuscript. The Lagrangian diapycnal overturning framework provides some unique insights into the overturning of the Subpolar North Atlantic. The authors also take this further by introducing the concept of a transformation efficiency, which they show to be useful because it is constant at leading order, meaning that the initial transport of the NAC is the main driver of variability. A common problem with new approaches like this is that they are difficult to compare with other studies in the literature. However, the authors successfully connect (or contrast) their results to prior results–most of which are Eulerian.

**I recommend this manuscript accepted in Ocean Science after the authors address the minor comments below.**

**Key Minor Comments:**
- In Figure 7 and L. 328-332: As far as I can tell, there is nothing from the analysis in this paper that attribute the diapycnal transformations to specific processes. More specifically, I don't see how you can infer that the RT waters are only mixing with ISOW and not also transformed by surface buoyancy fluxes. If this is an inferrence/speculation baked on prior analysis in Tooth et al. (2023) or other unrelated papers, then you should make that explicit.
- I find it confusing to call your main metric a "Lagrangian diapycnal overturning streamfunction" when it is not actually a streamfunction in the typical sense, but instead is just a difference between two Lagrangian transports. I find it much more appropriate to call it a "Lagrangian diapycnal transformation rate", which you do in some places.
- The specific conclusion that thermohaline anomalies are decoupled from diapycnal overturning seems to me to be a bit overblown. It is not obvious that the same result would apply to the Eulerian diapycnal overturning, which is what is measured by OSNAP and considered in conceptual models of the salt-advection feedback and AMOC stability.
- I encourage the authors to add more discussion of their Lagrangian experiment design, its caveats, and why they picked it over alternative approaches. If the goal was instead to understand the variability of NADW transported southwards across OSNAP West, then a

backtracking experiment would have been more appropriate, whereby particles are grouped according to their final time and convolved over many different release times. Would the authors expect to get the same qualitative results in that case?

**Minor line-by-line comments:**
- The acronym NAC is never defined!
- Can you explain why you use potential density referenced to the surface for the Lagrangian analysis but referenced to 2000 dbar for the Eulerian water mass analyses?
- Equation (4) seems incorrect to me. First, shouldn't the sum be over all density layers, since the box function is already picking out just the discrete outcropping layer? Second, I think you need to divide by the size of the density bin?
- L. 200-210 and Figure 3. Upon first read it is really hard to keep track of all of these transport numbers and how they are related. This is made even more difficult because the way you've rounded numbers means that things don't add up in a consistent way. For example, I was confused why the transports of 21.2 Sv and 3.9 Sv in Figure 3(b) did not add up to the total NAC transport 24.8 Sv. Can you round these up or down so that they're all self-consistent?
- Figure 3 panel labels are inconsistent with the description in the caption.
- Figure 3: Can you add the time-mean isopycnal that separates the two branches, $\sigma_{DWF}$?
- L. 208-210: Explain this comparison with observations better. Are the first two references some kind of analogous Lagrangian estimate of transports? Or an Eulerian transport but just for the strictly northward transport into the Nordic Seas whereas the 5.8 Sv (Østerhus) estimate is for the total transport?
- L. 222-225: The phrasing here is a bit confusing, especially because of the first sentence. I think what you mean is that: "Because 5.6 Sv of the water flowing northwards across OSNAP East is already in the lower limb, the 12.7 Sv of NADW formation is in fact a relatively larger fraction of the 19.2 Sv that is in the upper limb."
- L. 228: Can you cite a specific result from OSNAP here?
- L. 263: Worth emphasizing here (and perhaps in other places where it may be ambiguous) that the time t always refers to the time of "release", not the time at which transformation actually occurs or when it leaves across OSNAP West.
- L. 345 and Figure 8b-c. This is not a very interesting result and I think is mostly explained by the application of a binary sorting based on a fixed density threshold. Of course waters with inflow densities much less than $\sigma_{DWF}$ will need to transform more in order to cross the threshold. I would just cut these two panels.
- Figure 9b. What is the point of showing such a broad range of densities when we're only meant to focus on $\sigma_2=37.0$? Can you either plot this as a timeseries or zoom in on the denser waters a bit?

- Figure 9b Caption: Clarify that these are (I assume) anomalies relative to a monthly climatology.
- L. 403-408: This is really difficult to parse as written. I think what you mean is that, because dense waters do not outcrop in the western Labrador Sea, that suggests there is no significant formation of local NADW from waters coming north across the western part of OSNAP-W. But I don't really understand how that implies that \kappa < 100%.
- L. 479: "neither the efficiency of along-stream diapycnal transformation"...

---

## Author Comment (AC1)

**Controls on dense water formation along the path of the North Atlantic subpolar gyre**

Oliver J. Tooth, Helen L. Johnson
Chris Wilson

We are grateful to both of the reviewers for taking the time to read the manuscript and for providing constructive feedback. We have acted upon each of the suggestions proposed by the reviewers and we believe that these changes have significantly improved the clarity of our conclusions and the limitations of our findings.

Our response to Reviewers is structured as follows: our responses are included in **red** and the original Reviewer comments are included in **blue**.

**Responses to Reviewer 1 Comments**

**Key Minor Comments:**

**In Figure 7 and L. 328-332: As far as I can tell, there is nothing from the analysis in this paper that attribute the diapycnal transformations to specific processes. More specifically, I don't see how you can infer that the RT waters are only mixing with ISOW and not also transformed by surface buoyancy fluxes. If this is an inference / speculation baked on prior analysis in Tooth et al. (2023) or other unrelated papers, then you should make that explicit.**

The Reviewer is correct that we do not use the Lagrangian overturning framework directly to attribute along-stream diapycnal transformation to specific processes in this study. However, we are able to infer the character of the transformation (relative importance of surface-forced transformation versus localised diapycnal mixing) by examining the evolution of potential density, conservative temperature and absolute salinity along-stream. To be clear, we do not suggest that surface buoyancy forcing has no role to play in transforming SPG water parcels sourced from the subsurface of the Rockall Trough. Rather, we suggest that their downstream transformation is characteristic of intense diapycnal mixing with dense waters overflowing the Greenland-Scotland Ridge.

The importance of diapycnal mixing in the formation of lower NADW along the SPG pathway sourced from the Rockall Trough was deduced by examining the evolution of potential density along these water parcel trajectories, which clearly shows an abrupt densification consistent with localised diapycnal mixing south of the Greenland-Scotland Ridge (as previously found in Tooth et al. (2023)). We also note that Figure 7a shows that the SPG water parcels flowing northward in the Rockall Trough are mostly sub-surface (typically >

500 m depth). We have included below a reference Lagrangian probability map for an example release of Rockall-Trough origin water parcels (January 1995), alongside the evolution of their potential density downstream. This was generated by reproducing Figure 4a in the manuscript with only the subset of SPG water parcels which originate from the northward inflows to the Rockall Trough.

To avoid confusion for readers, we have removed the inferences ("Surface buoyancy loss + mixing along-stream" and "Mixing with Iceland-Scotland Overflow Water") from Figure 7c-d. We have also modified the text on Lines 340-342 to describe the character of NADW formation along SPG trajectories sourced from the Rockall Trough in terms of intense, localised diapcycnal transformation rather than diapycnal mixing, which is left for discussion on Lines 342-344.

[Figure]

**Figure R1** (a) Lagrangian probability map of a subset of SPG water parcels which flow northward across OSNAP East in the Rockall Trough in January 1995. (b) Evolution of potential density along the SPG pathway for the subset of water parcels which form NADW after flowing northward into the Rockall Trough in January 1995. The potential density sampled along each trajectory transiting from the NAC inflows across OSNAP East to 53N along OSNAP West is binned in discrete latitude-longitude space (Δx, Δy = 0.1 degrees) before calculating the average in each bin. The white

contour superimposed shows the location of the 27.66 kg m$^{-3}$ isopycnal used to distinguish between the AMOC upper and lower limbs.

**I find it confusing to call your main metric a "Lagrangian diapycnal overturning streamfunction" when it is not actually a stream function in the typical sense, but instead is just a difference between two Lagrangian transports. I find it much more appropriate to call it a "Lagrangian diapycnal transformation rate", which you do in some places.**

We are grateful to the Reviewer for highlighting this concern. We would firstly note that the Lagrangian diapycnal overturning stream function is not defined as the difference between two Lagrangian transports, but rather the accumulation of the net volume transport distribution in potential density coordinates from the lightest to the densest isopycnal surface. Thus, as highlighted previously in Tooth et al. (2023a), the Lagrangian overturning stream function is constructed analogously to an Eulerian overturning stream function, albeit with the volume transports calculated from a collection of water parcels rather than the integration of gridded velocities. Furthermore, when applied in the context of a trans-basin array (see Tooth et al. 2023a,b, 2024), we would expect the time-mean Lagrangian diapycnal overturning stream function to converge towards the equivalent Eulerian overturning stream function when averaged over a sufficiently long time-scale (recirculation time-scale of water parcel trajectories) - see p26 of Tooth (2024) for a detailed discussion.

We would also argue that naming our diagnostic the Lagrangian diapycnal overturning stream function makes for a more intuitive definition of North Atlantic Deep Water formation (Dense Water Formation: magnitude of the Lagrangian diapycnal overturning stream function at the time-mean isopycnal of maximum overturning, $\sigma_\theta$ = 27.66 kg m$^{-3}$) than using the "Lagrangian diapycnal transformation rate" (a term which is not included in the original manuscript).

We have therefore decided to keep the name "Lagrangian diapycnal overturning stream function" in the revised manuscript, since it maintains consistency with the existing literature on Lagrangian approaches to diagnosing water mass transformation (Döös et al., 2008; Kjellson & Döös, 2012; Berglund et al, 2021, 2022, 2023) and can be easily conceptualised as the total flux of water parcels into the lower limb.

Tooth, O. J. (2024). Lagrangian perspectives on the meridional overturning circulation in the North Atlantic Ocean [PhD thesis]. University of Oxford. https://ora.ox.ac.uk/objects/uuid:bd392502-8204-4dd3-b12d-ab88bd480794

Döös et al. (2008) https://doi.org/10.1029/2007JC004351

Kjellson & Döös., (2012) https://doi.org/10.1029/2012GL052420
Berglund et al. (2021) https://doi.org/10.1029/2021JC017330
Berglund et al. (2022) https://doi.org/10.1038/s41467-022-29607-8
Berglund et al. (2023) https://doi.org/10.1029/2022GL100989

**The specific conclusion that thermohaline anomalies are decoupled from diapycnal overturning seems to me to be a bit overblown. It is not obvious that the same result would apply to the Eulerian diapycnal overturning, which is what is measured by OSNAP and considered in conceptual models of the salt-advection feedback and AMOC stability.**

We are grateful to the Reviewer for raising this concern. We have modified the title of **Discussion** topic #2 to "Decoupling between upper limb thermohaline anomalies & subpolar dense water formation" to make it clearer to readers that our conclusion is that the amount of dense water formed along the boundary current of the SPG is independent of upper limb thermohaline anomalies (predominantly temperature anomalies on interannual-decadal timescales) flowing northward across OSNAP East. We have also modified the abstract on Lines 11-12 to be more precise; highlighting that we find a close relationship between the strength of the SPG and NADW formation on multi-decadal timescales. Moreover, we have revised the text on Lines 40-42, 399-400, 426-428 to remove possibly ambiguous references to "subpolar overturning" in favour of "subpolar dense water formation".

We agree with the Reviewer that the strong decoupling between upper limb thermohaline anomalies and downstream NADW formation may not project onto the subpolar overturning diagnosed in the Eulerian frame of reference. This potential decoupling between the Lagrangian DWF and the Eulerian diapycnal overturning measured at OSNAP is already discussed on Lines 263-269. We have also added to our discussion on Lines 450-452 to make it clear that our finding, that salinity anomalies can persist along the boundary current of the SPG, is compatible with the salt advection feedback, but is only likely to emerge as a control on subpolar dense water formation on longer, centennial timescales when salinity variability has been shown to play an influential role in determining low-frequency Eulerian overturning variability.

**I encourage the authors to add more discussion of their Lagrangian experiment design, its caveats, and why they picked it over alternative approaches. If the goal was instead to understand the variability of NADW transported southwards across OSNAP West, then a backtracking experiment would have been more appropriate, whereby particles are grouped according to their final time and convolved over many different release times. Would the authors expect to get the same qualitative results in that case?**

We thank the Reviewer for their comment and have made the following changes to the manuscript text:

- Updated Lines 134-136 in the **Model & Methods** section to make it clear to readers that evaluating Lagrangian trajectories forwards-in-time is essential to investigate the downstream impacts of thermohaline anomalies on dense water formation.
- On Lines 137-139, we highlight that this approach contrasts with previous work, which focuses on the sources (subduction locations) and pathways of NADW exported in the lower limb using backward-in-time trajectories.
- On Lines 142 - 144, we have included a further limitation of our experiment design: by using only a subsection of the OSNAP East array, we do not consider the small contribution made to the amount of dense water formed along the path of the SPG by upper limb water parcels flowing directly from the northernmost NAC branch to the Irminger Current.

The primary aim of our study is to investigate what governs the amount of dense water formed along the path of the SPG and whether thermohaline anomalies arriving in the eastern SPG have an impact. By definition, this requires us to advect water parcels forward-in-time to trace the evolution of thermohaline anomalies after flowing northward across OSNAP East. This also allowed us to relate the northward volume transport of upper limb waters to the magnitude of dense water formation. We would not be able to deduce this relationship using a backtracking experiment originating from the OSNAP West array (53N), given that the Lagrangian framework only permits us to constrain either the start (forward tracking) or end (backward tracking) time of our water parcels in exchange for conserving knowledge of their identity as they circulate around the subpolar North Atlantic.

Previous studies have used backward-in-time tracking of NADW flowing southward across OSNAP West to identify the sources (subduction locations; MacGilchrist et al., 2020; Fröhle et al., 2022) and export pathways (Georgiou et al., 2021). However, as highlighted by the Reviewer, attempting to establish whether the strength of NADW export across the OSNAP West array is related to upstream properties is confounded by the diversity of circulation pathways (e.g., Nordic Seas overflows and SPG-origin pathways) and water parcel transit times which are convolved in the Labrador Current. Our analysis of forward-in-time trajectories suggests that it would not be possible to trace temperature and salinity anomalies recorded along the OSNAP West array at 53N to a coherent thermohaline anomaly upstream in the eastern SPG.

Georgiou et al., (2021) https://doi.org/10.1029/2020JC016654
MacGilchrist et al., (2020) https://doi.org/10.1175/JCLI-D-20-0191.1
Fröhle et al., (2022) https://doi.org/10.5194/os-18-1431-2022

**Minor line-by-line comments:**

**The acronym NAC is never defined!**

We have modified the abstract to refer to the North Atlantic Current rather than the NAC and have defined the NAC acronym on Line 137 where the North Atlantic Current is first discussed in the main text.

**Can you explain why you use potential density referenced to the surface for the Lagrangian analysis but referenced to 2000 dbar for the Eulerian water mass analyses?**

We chose to use the potential density referenced to the sea surface for our Lagrangian analysis as this enables us to draw direct comparisons between the Lagrangian & Eulerian diapycnal overturning strength in the model and that observed along the OSNAP array (see Figures 1 and 4b). We would argue that decision is critical to ensure readers can understand our results in the context of previous literature on subpolar overturning dynamics (which generally use potential density reference to the sea surface).

With that said, we chose to perform the Eulerian surface-forced water mass transformation analysis using potential density referenced to 2000-m to closely follow the methodology used by Yeager et al. (2021) to define Labrador Sea Water in an ocean model. This approach is particularly favourable for two reasons: (1) it is more consistent with the mechanism of Labrador Sea Water formation by deep convection in both models and observations, and (2) it shares many similarities with the $\sigma_2$-based volumetric approach to Labrador Sea Water classification used in the observational studies of Yashayaev (2007a), Yashayaev et al. (2007b), Yashayaev et al. (2007c).

To make this justification clearer to readers, we have modified the text on Lines 186-188 to include:

 "*We use the potential density referenced to 2000 m, $\sigma_2$, in our Eulerian water mass transformation analysis to better identify Labrador Sea Water formation in the model, motivated by previous studies (e.g., Yashayaev, 2007a; Yashayaev et al., 2007b; Xu et al., 2018; Yeager et al., 2021).*"

Yashayaev (2007a) https://doi.org/10.1016/j.pocean.2007.04.015
Yashayaev et al. (2007b) https://doi.org/10.1029/2006GL028999
Yashayaev et al. (2007c) https://doi.org/10.1029/2007GL031812

**Equation (4) seems incorrect to me. First, shouldn't the sum be over all density layers, since the box function is already picking out just the discrete**

**outcropping layer? Second, I think you need to divide by the size of the density bin?**

We thank the Reviewer for highlighting the error in Equation 4. We have now modified the equation in line with the surface-forced water mass transformation equations presented in the studies of Petit et al. (2020), Yeager et al. (2021) and Megann et al. (2021).

Petit et al., (2020) https://doi.org/10.1029/2020GL091028
Yeager et al. (2021) https://doi.org/10.1126/sciadv.abh3592
Megann et al. (2021) https://doi.org/10.1029/2021JC017865

**L. 200-210 and Figure 3. Upon first read it is really hard to keep track of all of these transport numbers and how they are related. This is made even more difficult because the way you've rounded numbers means that things don't add up in a consistent way. For example, I was confused why the transports of 21.2 Sv and 3.9 Sv in Figure 3(b) did not add up to the total NAC transport 24.8 Sv. Can you round these up or down so that they're all self-consistent?**

**Figure 3 panel labels are inconsistent with the description in the caption.**

We thank the Reviewer for pointing out this typo and have now corrected the panel labels and updated the caption in Figure 3. On reviewing the breakdown of the total NAC transport (24.8 Sv), we found a minor error which led to the sum of the volume transport contributions of water parcels arriving in the Iceland Basin and Rockall Trough to be inconsistent with the total. We are grateful to the Reviewer for highlighting this and have now updated both Figure panels 3b-c and the accompanying text to ensure these are now self-consistent.

**Figure 3: Can you add the time-mean isopycnal that separates the two branches, \sigma_{DWF}?**

We are grateful to the Reviewer for this suggestion and have now included the time-mean position of both the 27.3 kg m$^{-3}$ and $\sigma_{DWF}$ = 27.66 kg m$^{-3}$ isopycnals. These specific isopycnals were chosen since they typically distinguish between the lighter (<= 27.3 kg m$^{-3}$) waters flowing northward across OSNAP East which continue to flow northward over the Greenland Scotland Ridge and those which recirculate in the SPG (> 27.3 kg m$^{-3}$ & < $\sigma_{DWF}$), and, in the case of $\sigma_{DWF}$, distinguish between the waters flowing northward in the time-mean upper limb and the lower limb of the subpolar AMOC as suggested by the Reviewer.

**L. 208-210: Explain this comparison with observations better. Are the first two references some kind of analogous Lagrangian estimate of transports? Or an Eulerian transport but just for the strictly northward transport into the Nordic Seas whereas the 5.8 Sv (Østerhus) estimate is for the total transport?**

We have improved this comparison on Lines 212-215 as suggested by the Reviewer to make it clear that the time-mean Lagrangian inflow transport across the Greenland-Scotland Ridge is larger than the observed Atlantic inflow to the Nordic Seas. However, since the time-mean strength of the Eulerian diapycnal overturning in the model and observations show a broad overall agreement, this implies that, in the model, a large fraction of the upper limb waters flowing northward across the Greenland-Scotland Ridge must be recirculated in the upper limb.

**L. 222-225: The phrasing here is a bit confusing, especially because of the first sentence. I think what you mean is that: "Because 5.6 Sv of the water flowing northwards across OSNAP East is already in the lower limb, the 12.7 Sv of NADW formation is in fact a relatively larger fraction of the 19.2 Sv that is in the upper limb."**

We are grateful to the Reviewer for highlighting this and have modified Lines 232-235 as suggested by the Reviewer to make our decomposition of the northward flow clearer to readers:

"*Of the 24.8 ± 4.2 Sv circulating around the SPG, Figure 4b indicates that 12.7 ± 1.9 Sv forms dense NADW (i.e., $\sigma_{53N}$ >= $\sigma_{DWF}$) prior to crossing OSNAP West. However, because 5.6 ± 1.4 Sv of the water flowing northwards across OSNAP East is already in the lower limb, this 12.7 ± 1.9 Sv of NADW formation represents a significant fraction of the 19.2 ± 3.0 Sv flowing northward across OSNAP East in the upper limb.*"

**L. 228: Can you cite a specific result from OSNAP here?**

Here we were highlighting that the results of our Lagrangian overturning analysis agree with the key result from the OSNAP observational programme: that water mass transformation in the eastern SPG (Iceland and Irminger Basins) dominates the total diapycnal overturning taking place in the SPG. To make this clearer to readers, we have modified the sentence on Lines 238-240 to:

"*In agreement with OSNAP observations (Lozier et al., 2019; Li et al., 2021), we find that the time-mean DWF$_{SPG}$ is dominated by NADW formation in the eastern SPG (9.0 ± 1.7 Sv; Fig. 4b).*"

**L. 263: Worth emphasizing here (and perhaps in other places where it may be ambiguous) that the time t always refers to the time of "release", not the time at which transformation actually occurs or when it leaves across OSNAP West.**

We thank the Reviewer for highlighting this potential source of confusion and have added the following sentence on Lines 274-276 to make this clearer:

*"We recall that the time t refers to the shared time when water parcels flow northward across OSNAP East, not the time at which they subsequently form NADW downstream."*

**L. 345 and Figure 8b-c. This is not a very interesting result and I think is mostly explained by the application of a binary sorting based on a fixed density threshold. Of course waters with inflow densities much less than \sigma_{DWF} will need to transform more in order to cross the threshold. I would just cut these two panels.**

On reflection, we agree with the Reviewer that the panels 8b-c did not add sufficient value beyond the existing Figure 8a to justify their inclusion and have removed these from the manuscript. We have also removed the accompanying text since the increasing homogeneity of water parcel properties on reaching OSNAP West (53N) is already demonstrated in Figure 7c-d and in the text on Lines 342-346.

**Figure 9b. What is the point of showing such a broad range of densities when we're only meant to focus on \sigma_{2}=37.0? Can you either plot this as a time series or zoom in on the denser waters a bit?**

**Figure 9b Caption: Clarify that these are (I assume) anomalies relative to a monthly climatology.**

We would like to thank the Reviewer for suggesting these valuable improvements to Figure 9b. We have now updated the panel to show the winter (DJFM) surface-forced water mass transformation anomalies relative to the 1975-2012 winter climatology for the narrower potential density range of 36.0 - 37.0 kg m$^{-3}$. Previously, we showed the annual mean surface-forced water mass transformation anomalies relative to the 1975-2012 climatology; however, on reflection, we believe isolating the wintertime transformation is more informative given the strongly seasonal nature of subpolar diapycnal transformation.

We have also modified the Figure 9 caption to make this obvious to readers:

*"Winter (DJFM) surface-forced Water Mass Transformation (WMT) anomalies relative to the 1975-2012 winter climatology calculated over the Lagrangian experiment domain (see Fig. 1a) in $\sigma_2$ potential density coordinates (referenced to 2000 m)."*

**L. 403-408: This is really difficult to parse as written. I think what you mean is that, because dense waters do not outcrop in the western Labrador Sea, that suggests there is no significant formation of local NADW from waters coming**

**north across the western part of OSNAP-W. But I don't really understand how that implies that \kappa < 100%.**

We recognise that the original text was insufficiently clear here and have removed reference to outcropping isopycnals in the western Labrador Sea. Instead, we now simply remark that the presence of both upper and lower limb waters flowing southward in the Labrador Current implies that "*, in reality, the SPG circulation projects onto a diapycnal overturning cell (and thus the formation of NADW) with a time-evolving efficiency characterised by $\kappa(t) <$ 100%.*"

**L. 479: "neither the efficiency of along-stream diapycnal transformation"...**

We have updated the text as suggested by the Reviewer on Line 480.

---

## Author Comment (AC2)

**Controls on dense water formation along the path of the North Atlantic subpolar gyre**

Oliver J. Tooth, Helen L. Johnson
Chris Wilson

We are grateful to both of the reviewers for taking the time to read the manuscript and for providing constructive feedback. We have acted upon each of the suggestions proposed by the reviewers and we believe that these changes have significantly improved the clarity of our conclusions and the limitations of our findings.

Our response to Reviewers is structured as follows: our responses are included in **red** and the original Reviewer comments are included in **blue**.

**Responses to Reviewer 2 Comments**

**Why is the focus of this paper on dense water formation rather than the AMOC explicitly? According to the findings from OSNAP, there is no connection between dense water formation and dense water export (i.e., AMOC), see Zou and Lozier (2016).**

We make it clear to readers from the outset that the focus of our study is on dense water formation rather than the AMOC (this is also reflected in the title of the manuscript). This choice was made for two reasons:

1. Our Lagrangian experiment seeks to quantify the formation of NADW by only one component of the subpolar overturning circulation: the boundary current of the SPG. We thus do not quantify NADW formation due to upper limb water parcels transformed in the Nordic Seas and the Arctic Ocean or those flowing directly into the Irminger Sea via the northernmost branch of the North Atlantic Current. To focus our study on the basin-scale overturning in the subpolar North Atlantic, we would need to account for all of these components. This limitation is included on Line 243, where we highlight that the Lagrangian diapycnal overturning strength in the eastern SPG cannot be compared to the traditional Eulerian diapycnal overturning strength at OSNAP East, given that we do not include the contribution of the Nordic Seas overflows.

2. As discussed on Lines 263-269, we would not necessarily expect dense water formation along Lagrangian trajectories circulating around the SPG to imprint onto the Eulerian diapycnal overturning (AMOC) strength. This is because water parcels will enter the lower limb along the entire length of the SPG boundary current, such that the time taken for newly formed NADW to reach OSNAP West and imprint onto the subpolar AMOC strength could vary from days to years. Furthermore, given that we do not continue to track water parcels following their southward crossing of OSNAP West (53N), it is not possible to distinguish between the newly formed NADW

parcels which are exported from the SPNA (thereby contributing to the basin-scale AMOC) from those which are simply recirculated within the SPG. This challenge is highlighted as a topic for further research in our Discussion on Lines 478-481.

Finally, the Reviewer is correct in highlighting that OSNAP observations show a weak (rather than non-existent) relationship between deep convection in the interior of the Labrador and Irminger Seas and the strength of the Eulerian overturning recorded along each array. However, our Lagrangian analysis focuses on the water parcels which are both transformed and exported within the boundary current of the SPG (see Figure 4a) rather than those experiencing wintertime convection in the basin interior. We would also highlight that both Le Bras et al. (2020) and Li et al. (2021) found a much stronger relationship between seasonal water mass transformation (convection) taking place within the boundary current and the downstream export of UNADW across OSNAP East.

Le Bras et al. (2020) https://doi.org/10.1029/2019GL085989
Li et al. (2021) https://doi.org/10.1038/s41467-021-23350-2

**The authors should explain how their results impact the idea that temperature or salinity anomalies propagate on a steady ocean circulation (e.g. Sutton and Allen, 1997; Årthun et al., 2017), rather than a varying ocean circulation creates temperature and salinity anomalies (e.g. Foukal and Lozier, 2016; Desbruyères and Chafik, 2021).**

We appreciate the Reviewer's suggestion, but note that we do not explicitly consider the origins of temperature and salinity anomalies arriving in the eastern SPG in our study. Rather, we are concerned with their downstream consequences for dense water formation: our results suggest that temperature anomalies are damped along the boundary current of the SPG, while salinity anomalies can persist downstream to impact dense water formation.

In the model, we do find a relationship between upper ocean temperature and salinity anomalies and the state of the subpolar circulation: a weaker, slower SPG circulation (see Fig. 9f) is associated with warmer, lighter upper limb waters (see Fig. 10b) flowing northward across OSNAP East. However, diagnosing the source of the thermohaline anomalies arriving in the eastern SPG is not in the scope of the present study since it would require us to evaluate backward-in-time Lagrangian trajectories to identify changes in the sources of the northward flowing waters arriving at OSNAP East. Such analyses have already been performed by Fox et al. (2022), Foukal and Lozier (2016), and Desbruyeres et al., (2021) as highlighted by the Reviewer, and emphasise that the composition of subtropical- vs. subpolar-origin waters arriving in the eastern SPG acts as an important control on upper ocean properties.

**Fig. 1: the model streamfunction is also broader than observations, which indicates that the upper limb waters are lighter than observed and the water mass transformation in the subpolar North Atlantic is larger than observed. Furthermore, there is considerably more formation of very dense waters (sigma>27.75 kg/m3), which implies that this model suffers from the well-known issue of too strong convection in the Labrador Sea (e.g., Menary et al., 2020). This issue should be discussed in the conclusions as a limitation of the study.**

We thank the Reviewer for highlighting model biases in the magnitude and composition of diapycnal transformation as a relevant limitation of our study. We have now added a short paragraph on Lines 483-487 of the Discussion highlighting the potential relevance of such biases to our conclusions:

**"***We also recognise that model biases may play a role in amplifying the relationship between remote surface buoyancy forcing and DWF along the path of the SPG in this ocean model. For example, the larger-than-observed lower NADW formation (> 27.75 kg m$^{-3}$ in Figure 1b) north of OSNAP West in this hindcast is indicative of excessive Labrador Sea deep convection (a well-established bias in eddy-rich models; Petit et al., 2023, Jackson and Petit, 2023), which would enable the deeper penetration and greater persistence of density anomalies originating from surface buoyancy forcing (Reintges et al., 2024).***"**

**Line 100: OSNAP imposes a -1.6 Sv flow through OSNAP West and a +1.6 Sv flow through OSNAP East (Lozier et al., 2019). Did the authors consider the effect of this northward flow across OSNAP East as well?**

We already comment on the influence of the weaker than observed net throughflow in the model on the strength of diapycnal overturning at OSNAP East on Lines 95-97: "*... However, this is primarily due to the weaker time-mean net northward transport across the section in the model (0.8 ± 1.1 Sv) compared to the 1.6 Sv imposed in the OSNAP observational calculation.*" If we were to impose a similar 1.6 Sv net throughflow across OSNAP East in the model, the resulting time-mean overturning strength would be approximately 14.3 Sv (13.5 + 1.6 Sv - 0.8 Sv) and hence would compare even more favourably with the 14.5 Sv observed along the OSNAP array. Importantly, our Lagrangian overturning calculations are not influenced by the water parcels which contribute to the net northward transport across the OSNAP East section, given that we remove these water parcels from our experiment on their northward crossing of the Greenland-Scotland Ridge.

**Fig. 2: How does the strength of the 'SPG pathway' compare to a Eulerian measure of SPG strength, such as from OSNAP?**

We thank the Reviewer for this interesting question. Unfortunately, there is no simple approach to compare the volume transport of an individual Lagrangian circulation pathway, such as the SPG boundary current, to an Eulerian measure of the SPG strength. This is because integrated Eulerian diagnostics, such as the barotropic stream function, will also include the volume transport contributions of the Irminger Gyre, Arctic-origin and Nordic Seas overflow pathways. We highlight the challenge of comparing traditional Eulerian metrics with Lagrangian diagnostics describing an individual component of the flow on Lines 243-244, where we note that the DWF occurring in the eastern SPG cannot be directly compared to the Eulerian overturning stream function calculated at OSNAP East due to the absence of the Nordic Seas overflows in our Lagrangian analysis.

**Fig. 3: This is a beautiful figure – please overlay isopycnals on panels b and d to look at baroclinicity in the water column. The strength of the baroclinicity in different parts of the**

**region could explain why some water masses make it over the Greenland-Scotland Ridge and some are retained in the subpolar basin.**

We thank the Reviewer for their excellent suggestion to improve Figure 3. We have now included the time-mean position of the both the 27.3 kg m$^{-3}$ and $\sigma_{DWF}$ = 27.66 kg m$^{-3}$ isopycnals. These specific isopycnals were chosen since they typically distinguish between the lighter (<= 27.3 kg m$^{-3}$) waters flowing northward across OSNAP East which continue to flow northward over the Greenland Scotland Ridge and those which recirculate in the SPG (> 27.3 kg m$^{-3}$ & < $\sigma_{DWF}$), and, in the case of $\sigma_{DWF}$, distinguish between the waters flowing northward in the time-mean upper limb and the lower limb of the subpolar AMOC in this model.

**Line 365: Hakkinen and Rhines (2004) used an EOF of SSH to derive their 'gyre index', not a SSH gradient as indicated in the text here. See Foukal and Lozier (2017) for a discussion of the 'gyre index' in comparison to a SSH gradient metric. See also Chafik and Lozier (2025) for further discussion of why the gyre index is not a good metric of subtropical-to-subpolar connectivity.**

We are grateful to the Reviewer for highlighting this inconsistency between the Hakkinen and Rhines (2004) reference and the text. Our intention here was to recognise the work of Hakkinen and Rhines (2004) in demonstrating the relationship between sea surface height and the subpolar gyre strength (specifically, that a weakening of the the subpolar gyre circulation is associated with an increase in sea surface height as captured by the 1st temporal mode - PC1). On reflection, however, we believe that the work of Yeager et al. (2020) demonstrating how abyssal thickness anomalies induce changes in the sea surface height gradient across the basin (and hence modify the near-surface geostrophic velocity field) is more relevant to our findings and have now included this in place of the Hakkinen and Rhines (2004) reference on Lines 370-372.

Yeager et al. (2020) https://doi.org/10.1007/s00382-020-05382-4

**Line 384-386: This paragraph should include the context that this relationship occurs in the model they are analyzing, and may not apply to the real ocean. The authors should consider adding this caveat to other parts of their paper as well.**

We agree with the Reviewer that our summary should have been more precise regarding the source of our conclusions. We have now added "*in an eddy-rich ocean model hindcast*" to the end of Lines 390-392 to make this clearer to readers.

We have also added a similar caveat to the summary on Lines 426-429 in the Discussion:

"*... indicating that upper limb potential density anomalies do not feed back onto the strength of DWF and hence diapycnal overturning in this eddy-rich ocean model.*"

And on Line 496:

*"Instead, decadal variations in the DWF along the path of the SPG are driven remotely by surface buoyancy forcing localised in the central Labrador and Irminger Seas in this model."*

**Figure 1 appears before its first mention (line 90).**

We have moved Figure 1 to be positioned at the end of Methods section 2.1 (above Line 125), following its first reference in the text on Line 91, as suggested by the Reviewer.